# A reconfigurable and magnetically responsive assembly for dynamic solar steam generation

Yajie Hu [1], Hongyun Ma [1], Mingmao Wu[1], Tengyu Lin[1,2], Houze Yao[1], Feng Liu [3] ✉, Huhu Cheng [1] ✉ & Liangti Qu [1] ✉

Interfacial solar vapor generation is a promising technique to efficiently get fresh water from seawater or effluent. However, for the traditional static evaporation models, further performance improvement has encountered bottlenecks due to the lack of dynamic management and self-regulation on the evolving water movement and phase change in the evaporation process. Here, a reconfigurable and magnetically responsive evaporator with conic arrays is developed through the controllable and reversible assembly of graphene wrapped $Fe_3O_4$ nanoparticles. Different from the traditional structure-rigid evaporation architecture, the deformable and dynamic assemblies could reconfigure themselves both at macroscopic and microscopic scales in response to the variable magnetic field. Thus, the internal water transportation and external vapor diffusion are greatly promoted simultaneously, leading to a 23% higher evaporation rate than that of static counterparts. Further, well-designed hierarchical assembly and dynamic evaporation system can boost the evaporation rate to a record high level of 5.9 kg m$^{-2}$ h$^{-1}$. This proof-of-concept work demonstrates a new direction for development of high performance water evaporation system with the ability of dynamic reconfiguration and reassembly.

Water shortage is becoming a worldwide ecological challenge to human beings. Over two-thirds of people in the world are facing water scarcity[1]. Interfacial solar steam generation (ISSG) is deemed to be a promising solution to get freshwater from seawater or wastewater[2]. Compared with traditional solar distillation, ISSG cuts down the heat loss via localizing the solar heat near the surface of the evaporator. Up to now, numerous materials including metal nanoparticles[3–5], carbonaceous materials[6–8], two-dimensional (2D) materials[7,9–11], hydrogels[12–14] were explored as evaporators. Meanwhile, many effective methods have also been proposed to enhance the evaporation rate, concerning three-dimensional (3D) ISSG[15–20], multistage evaporation[21–24], water supply control[25,26], synergy with chemical phase change[27–29] and electrothermal effect[30,31], reduction of evaporation enthalpy[12,13,32] and so on. However, further improvement of evaporation performance has encountered bottlenecks[33,34]. The conventional evaporators are structurally fixed, incapable of disassembling and deforming, thus generally inducing the salt deposition and lowering the evaporation efficiency[35–38], although different strategies have been tried including Janus structure[18,36], localized crystallization[35], contact-less evaporation[37,39,40], convective flow of water[38,41], and wick-free water layer[42]. Most of them only involve the passive fluidic flow of water, the active circulation of water and salt ions has not been reported, which could deal with the problem of the slow diffusivity in passive convection[42]. On the other hand, in these static evaporation systems,

[1]Key Laboratory of Organic Optoelectronics & Molecular Engineering, Ministry of Education, Department of Chemistry & State Key Laboratory of Tribology, Department of Mechanical Engineering, Tsinghua University, Beijing 100084, People's Republic of China. [2]HurRain Nano Technology Co., Ltd, Beijing 100084, People's Republic of China. [3]State Key Laboratory of Nonlinear Mechanics, Institute of Mechanics, Chinese Academy of Sciences, Beijing 100190, People's Republic of China. ✉e-mail: liufeng@imech.ac.cn; huhucheng@tsinghua.edu.cn; lqu@mail.tsinghua.edu.cn

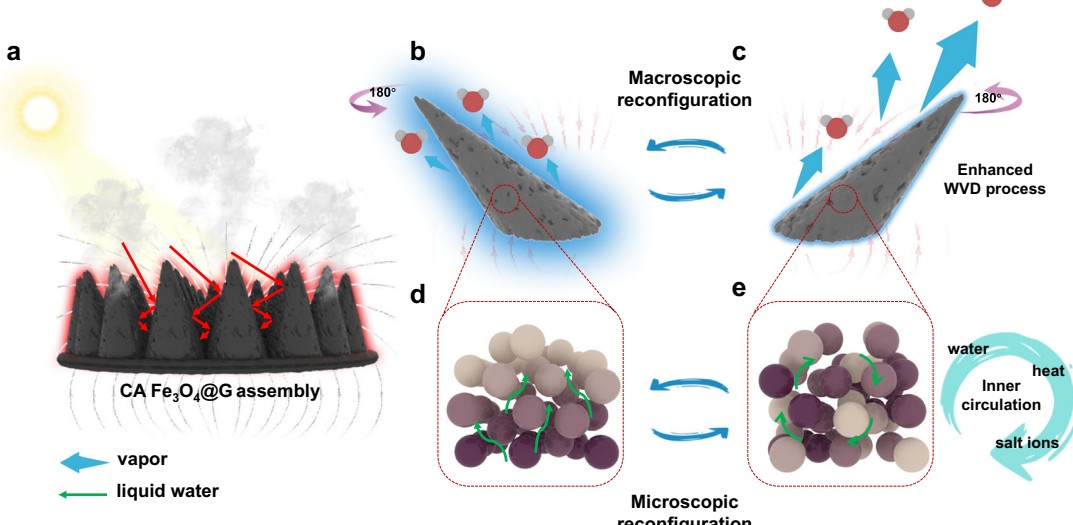

**Fig. 1 | The concept of dynamic evaporation enabled by reconfigurable Fe₃O₄@G assembly. a** Schematic diagram of the dynamic solar water evaporation with conic arrays (CAs) consisted of graphene wrapped Fe₃O₄ (Fe₃O₄@G) nanoparticles. **b, c** Macroscopic reconfiguration of an individual CA assembly during dynamic evaporation. In response to the external magnetic field, the CA assembly deforms itself from the left (**b**) to the right tilt state (**c**), during which the disturbance to the water vapor in a near circumstance results in the enhancement to water vapor diffusion (WVD) process. **d, e** Microscopic reconfiguration of Fe₃O₄@G nanoparticles during dynamic evaporation. With the CA assembly changing orientation, the inside nanoparticles keep moving from the initial state (**d**) to the disordered state (**e**), the nanoparticles are colored with different color depth to show the relative positions. Without reconfiguration, water and salt ions could only undergo unidirectional transport, while microscopic reconfiguration establishes an inner circulation of water, salt ions, and heat.

evaporation rate is largely restricted by the accumulated water vapor around the evaporation surface, which results from the insufficient water vapor diffusion (WVD) from evaporators to surroundings[43,44]. As for this point, pioneer work has been conducted to enhance WVD process with airflow[43], while the forced convective flow is of low efficiency and incompatible with the current closed apparatus. Therefore, promoting WVD is the key for further improvement of evaporation rate, which is still challenging to solve to date.

Here, we presented a dynamic and reconfigurable solar evaporator via the reversible assembly of graphene wrapped Fe₃O₄ (Fe₃O₄@G) nanoparticles. As illustrated in Fig. 1a, the unique conic array (CA) assembly of Fe₃O₄@G nanoparticles is constructed by Rosensweig instability under magnetic field and can deform into various shapes on demand. The well-organized CA assembly possesses excellent salt resistant features, recycling ability as well as high evaporation rate. The real-time reconfiguration of CA assembly in a variable magnetic field renders a macroscopical disturbance to the surrounding atmosphere, greatly promoting WVD (Fig. 1b, c). Meanwhile, on a microscopic level, the internal nanoparticles adaptively rearrange themselves within CA assembly, establishing an inner circulation of substances for the favorable water transportation (Fig. 1d, e). As a result, the 23% enhancement of evaporation rate is achieved compared with static evaporation. As we shall see later, based on a three-dimensional magnetic hierarchical assembly of Fe₃O₄@G, the static evaporation rate is improved up to 4.80 kg m⁻² h⁻¹ at one solar illumination of 1 kW m⁻², and could even be further boosted to record high level of 5.88 kg m⁻² h⁻¹ in a dynamic evaporation process. This proof-of-concept work provides a new angle of view toward the development of dynamically adjustable advanced solar water evaporation systems.

## Results

### Preparation and characterization of Fe₃O₄@G nanoparticles
Fe₃O₄@G nanoparticles are prepared through a facile two-step process. As illustrated in Fig. 2a, graphene oxide wrapped Fe₃O₄ (Fe₃O₄@GO) nanoparticles are firstly constructed based on the electrostatic attraction between Fe₃O₄ nanoparticles and graphene oxide

(GO), then Fe₃O₄@G nanoparticles are obtained by the reduction of GO via thermal annealing (see Methods for details). The as-prepared Fe₃O₄@G nanoparticle has a uniform graphene shell consisting of a few layers of graphene sheets (Fig. 2b). The layer spacing of the external shell is 0.34 nm, corresponding to the van der Waals spacing of graphite, and the lattice fringes of the internal core are in line with the crystal plane of Fe₃O₄, indicating the neat core-shell structure of Fe₃O₄@G (Fig. 2c). Owing to the steric hindrance and lubrication effect of graphene shells, Fe₃O₄@G nanoparticles show an even distribution without aggregation (Fig. 2d). In detail, Zeta potential (Fig. 2e) confirms the electrostatic interaction between positively charged Fe₃O₄ nanoparticles and negatively charged GO sheets, as well as their successful combination indicated by the mediate Zeta potential. Raman spectrum was investigated to track the change of the graphene shell at each synthetic stage (Fig. 2f). Compared with bare Fe₃O₄ nanoparticles, the existence of graphene (oxide) is confirmed in Fe₃O₄@GO and Fe₃O₄@G as the typical D peak and G peak emerge at 1350 and 1580 cm⁻¹, respectively. And the decrease of $I_D/I_G$ ratio from 2.93 to 2.01 further proves the reduction from GO to graphene. Besides, Fe₃O₄ remains unchanged without oxidation during the synthetic process as confirmed by X-ray differential (XRD) spectrum (Fig. 2g). Thermal gravimetric analysis (TGA) (Fig. 2h) shows that bare Fe₃O₄ nanoparticles are oxidized as temperature increases (green region). The same trend also exists in the profiles of Fe₃O₄@GO and Fe₃O₄@G, but the reduction of GO leads to the deviation of Fe₃O₄@GO before 300 °C (red region). Meanwhile, the excessive oxidation at high temperature removes graphene away from the system, causing the weight losses above 350 °C (blue region). Benefiting from the uniform but thin shell of graphene, the magnetic properties of Fe₃O₄@G are well maintained (Fig. 2i), ensuring the excellent response to the magnetic field (Supplementary Fig. 2). At the same time, the absorption of sunlight is enhanced both at visible and infrared bands, the overall sunlight absorption is increased up to 98.4% (Fig. 2j).

### Reconfiguration ability of Fe₃O₄@G assembly
Benefiting from the graphene wrapping core-shell structure, Fe₃O₄@G nanoparticles possess remarkable characteristics of reversible

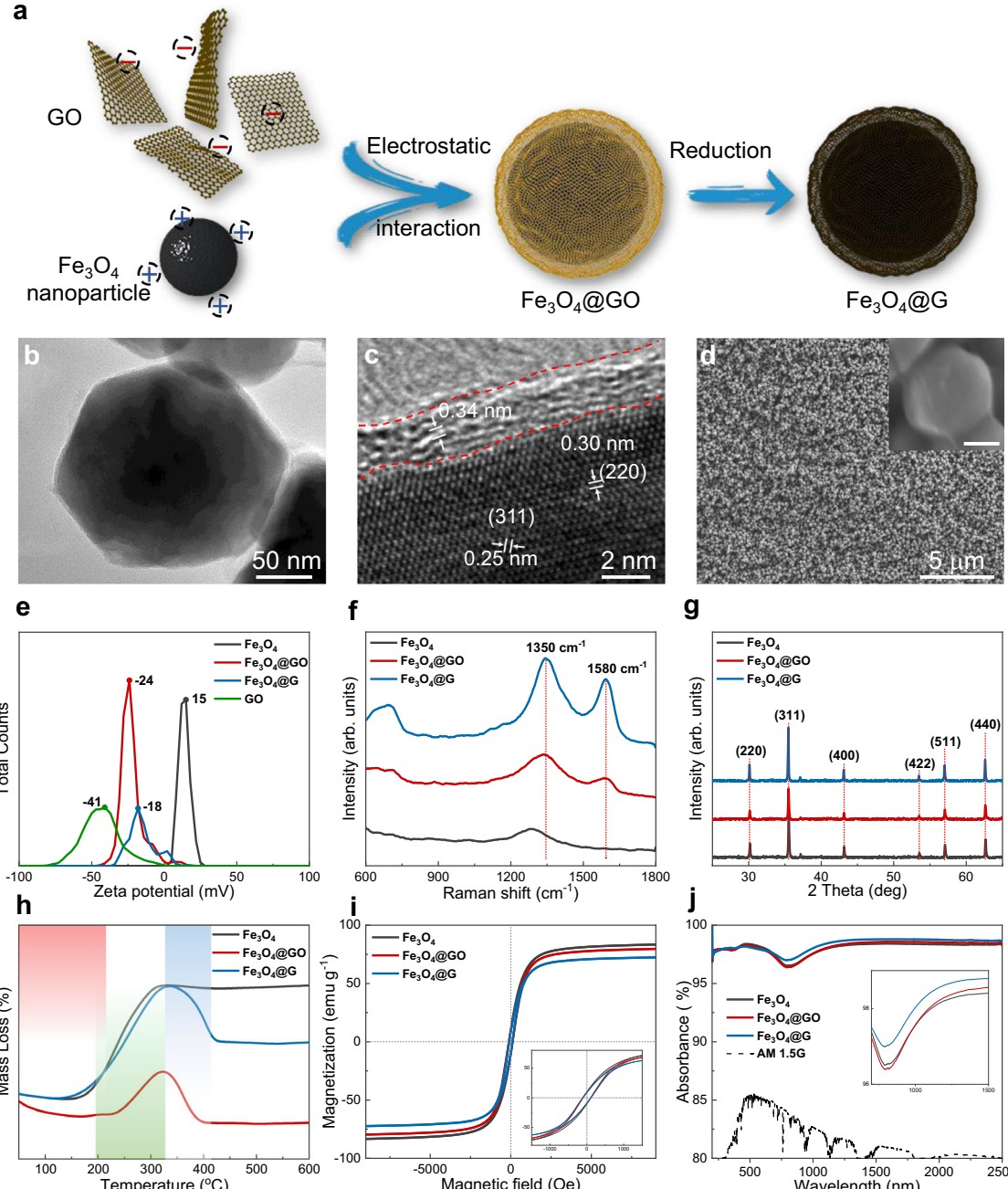

**Fig. 2 | The preparation and characterization of Fe₃O₄@G nanoparticles.**
**a** Schematic diagram for the facile two-step synthesis of Fe₃O₄@G.
**b**, **c** Transmission electron microscopy (TEM) images of Fe₃O₄@G nanoparticle at different magnification. **d** Scanning electron microscopy (SEM) images of evenly distributed Fe₃O₄@G nanoparticles at low magnification. Inset image is the high-resolution SEM image showing the uniform wrapping of graphene sheets on Fe₃O₄@G nanoparticle. Scale bar, 100 nm. **e** Zeta potential of graphene oxide (GO), Fe₃O₄, graphene oxide wrapped Fe₃O₄ (Fe₃O₄@GO), and Fe₃O₄@G. **f** Raman

spectra of Fe₃O₄, Fe₃O₄@GO, and Fe₃O₄@G. **g** XRD spectra of Fe₃O₄, Fe₃O₄@GO, and Fe₃O₄@G. **h** TGA curves of Fe₃O₄, Fe₃O₄@GO, and Fe₃O₄@G. The red region, green region, and blue region refer to the periods of reduction of GO, oxidation of Fe₃O₄, and excessive oxidation of graphene, respectively. **i** Magnetic hysteresis loops of Fe₃O₄, Fe₃O₄@GO, and Fe₃O₄@G. Inset is the magnified figure of the hysteresis loops. **j** Ultraviolet-Visible-Near infrared spectra of Fe₃O₄, Fe₃O₄@GO, Fe₃O₄@G, and sunlight. Inset image is the enlarged figure at visible and infrared band. Source data are provided as a Source Data file.

assembly and reconfiguration. A typical reconfiguration process of Fe₃O₄@G assembly is exemplified in Fig. 3a–c. Firstly, a free-standing CA assembly could be constructed beforehand on the left panel (Fig. 3a). With controlling the magnetic field and water content, CA assembly could be immediately disassembled into suspension of Fe₃O₄@G nanoparticles, which can deform itself to pass through a narrow valley from left to right side (Fig. 3b), indicating its excellent deformability. Finally, a regenerated CA assembly could be secondly constructed at the right panel (Fig. 3c). The reconfiguration ability of

Fe₃O₄@G assembly was achieved owing to the following factors: (i) irreversible aggregation was avoided from the steric hindrance and lubrication effect of graphene[45], (ii) a hierarchical porous structure was constructed resulting from the rational regulation of surface hydrophilicity, (iii) thus endowing the assemblies of Fe₃O₄@G with disassembly ability as well as the excellent water transportation ability. Compared with bare Fe₃O₄ nanoparticles, no irreversible aggregation of Fe₃O₄@G was observed after multiple recycling, while large particles formed among bare Fe₃O₄ nanoparticles due to the Ostwald

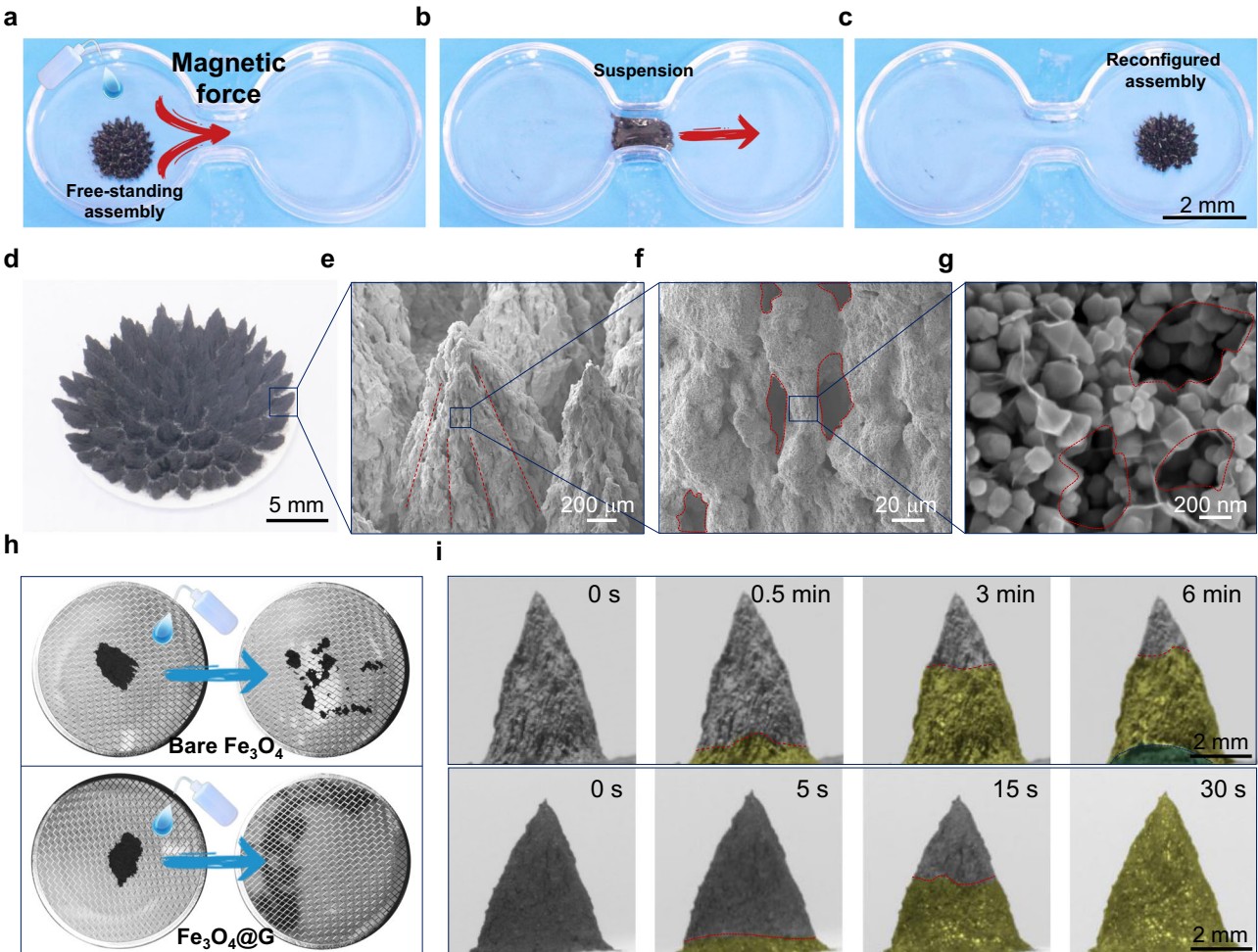

**Fig. 3 | The reconfiguration ability of Fe₃O₄@G assembly. a–c** Manifestation for the reconfiguration process of Fe₃O₄@G assembly. The conic array (CA) assembly was immediately disassembled upon withdrawing magnetic field (**a**), then was transferred by magnetic force to pass through a narrow valley to the other side (**b**), and finally was reconfigured to a new one under the control of magnetic field (**c**). **d** The optical image of CA assembly. **e–g** Consecutive SEM images of CA assembly with different magnification, showing that each CA assembly has oriental grooves from bottom up (**e**), and pores with diameters ranging from 20 μm (**f**) to 100 nm (**g**). **h** Comparison of the disassembly process between bare Fe₃O₄ and Fe₃O₄@G. Only Fe₃O₄@G assembly can fully disassemble and pass through the sieve. **i** Comparison of bottom-up water transport processes between the CA assemblies of bare Fe₃O₄ and Fe₃O₄@G. The top panel and bottom panel correspond to bare Fe₃O₄ assembly and Fe₃O₄@G assembly, respectively. The red dashed line indicates the water frontier, yellow regions are the water infiltrated part, and the blue region is the liquid water.

ripening effect[46] and irreversible aggregation[47] (Supplementary Fig. 9). Additionally, a hierarchical porous structure was formed in the assemblies of Fe₃O₄@G. As depicted in Fig. 3d, the CA assembly has a periodic pattern consisting of several individual conic assemblies, in which oriented grooves taper from bottom up, leading to the rapid transportation of water (Fig. 3e). Consecutive SEM images in Fig. 3f, g demonstrate the multi-scale pores with diameters ranging from several millimeters to hundreds of nanometers, which simultaneously ensures the disassembly ability as well as the adequate water supply. The detailed mechanism was elucidated in Supplementary Figs. 10–12 for the formation of the hierarchical porous structure. The bare Fe₃O₄ nanoparticles tend to agglomerate into a compact and rigid structure, which hinders the disassembly process and water transport. Compared with bare Fe₃O₄ nanoparticles, the capillary force was controlled slightly lower by graphene wrapping (Supplementary Fig. 10), thus decreasing the attraction between nanoparticles and resulting in the generation of pores and disassembly ability. As shown in Fig. 3h, the assembly made of Fe₃O₄@G could be easily taken apart and pass through the microporous sieve, while that of the bare Fe₃O₄ nanoparticles failed to disassemble (Supplementary Movie 1). Meanwhile, the water transportation process could be monitored by camera owing

to the variation of the refractive index of light (Fig. 3i). For bare Fe₃O₄ assembly, water cannot reach the top of the CA assembly even after 6 min and the water transport rate is limited to 8 μm s⁻¹. By contrast, for Fe₃O₄@G assembly, water was rapidly absorbed and transported to the tip of the CA assembly within 30 s, and the water transport rate is up to 167 μm s⁻¹ which is two orders of magnitude higher than that of bare Fe₃O₄ assembly, indicating sufficient water supply for the evaporation.

### Static evaporation performance of the CA assembly
The static evaporation of the CA assemblies was studied in a traditional apparatus shown in Fig. 4a. The assembly was located on a glass fiber substrate, which hardly absorbs sunlight as confirmed in Supplementary Fig. 16. The polystyrene foam was used to cut down heat loss, and cotton swab was employed as a water path. Assemblies can be fabricated into different shapes owing to the reconfiguration ability of Fe₃O₄@G nanoparticles (Supplementary Figs. 17 and 18). Among them, CA assembly has the highest evaporation rate, which is as high as 1.77 kg m⁻² h⁻¹ and 8% higher than that of the planar counterparts (Fig. 4b). More importantly, CA assembly intrinsically has great salt-resistant ability. For evaporation of the simulated seawater with

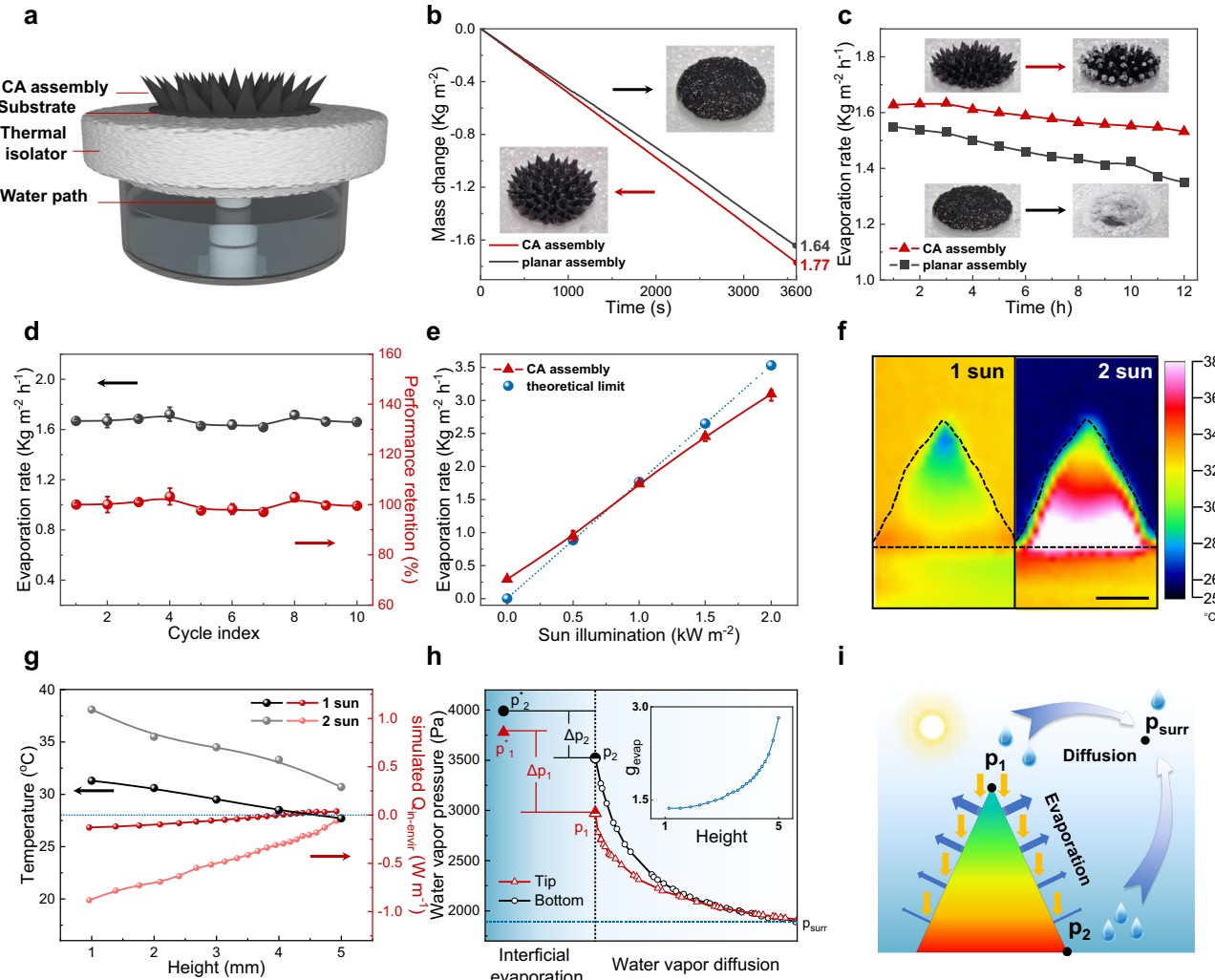

**Fig. 4 | Static evaporation performance of the CA assembly. a** Schematic diagram of the evaporation apparatus. **b** Mass change of the conic array (CA) vs. planar assemblies under 1 sun illumination. Insets are the corresponding optical images. **c** Twelve-hour long-term evaporation test with 3.5 wt% seawater. Inset images demonstrate the salt precipitation after test, the upper is for CA assembly and the lower is for planar assembly. **d** Recycling evaporation test of CA assembly with 3.5 wt% salt water ($n = 3$, error bars: standard deviation). **e** Evaporation rate of CA assembly under different illumination intensity. The blue dashed line indicates the theoretical limit ($n = 3$, error bars: standard deviation). **f** Infrared images of CA assembly under 1 and 2 sun illumination during static evaporation. Scale bar, 2 mm.

**g** Plots of surface temperature and the simulated $Q_{in\text{-}envir}$ of a CA assembly with respect to height under 1 and 2 sun illumination. The blue dashed line indicates the environmental temperature where no energy exchange with the environment. **h** Plots of simulated water vapor pressure from evaporation surface of the top and the bottom to the surroundings, inset is the plot of the simulated evaporation rate with respect to the height under 1 sun illumination. The horizontal axis unit, mm; the vertical axis unit, $10^{-4}$ kg m$^{-2}$ s$^{-1}$. **i** Schematic image for the evaporation model. $p_1$, $p_2$, and $p_{surr}$ are the water vapor pressures of the top and bottom position of the CA assembly and of the surroundings, respectively. Source data are provided as a Source Data file.

salinity of 0.8–20 wt%, evaporation rates of CA assembly remain higher than that of planar one (Supplementary Fig. 19). In a 12 h evaporation test (Fig. 4c), the salts only precipitate on the tips of CA assembly and hardly interfere with the incident sunlight[35]. After 12 h evaporation, the attenuation of evaporation rate was only 5%. On the contrary, the planar one was entirely covered by the precipitated salt, accompanying with a large decrease in evaporation rate of 12%. Besides, thanks to the efficient water transport in CA assembly, a self-dissolving process could take place at dark, and almost all the formed salts can be removed within hours (Supplementary Fig. 20). Furthermore, in case of undissolved salts, CA assembly can fully regenerate itself via a reconfiguration process. As illustrated in Fig. 4d, no degradation of evaporation performance was found after 10 times recycling assembly of Fe$_3$O$_4$@G, and all the salts were removed after regeneration as confirmed by the EDS mapping shown in Supplementary Figs. 23 and 24. The reconfiguration on the level of nanoparticles endowed CA assembly with excellent durability for sustainable evaporation.

To understand the enhancement effect of CA assembly and seek for further improvement, evaporation rates at different sunlight illumination were tested (Fig. 4e). Under illumination of 0–1000 W m$^{-2}$, the evaporation rate of CA assembly is higher than the theoretical limit, which is calculated from the evaporation enthalpy[12,13,48] (Supplementary Fig. 21). However, it becomes lower than the theoretical limit when illumination larger than 1000 W m$^{-2}$. In order to figure out the heat flow directions, real-time infrared thermal imaging is applied. As confirmed in Fig. 4f, a gradient surface temperature distribution exists over the CA assembly, where the temperature is higher at the bottom and lower at the tip. For 1 sun illumination, the temperature at the tip is 27.7 °C, which is a bit lower than the environment temperature of 28 °C, indicating that heat flows from the environment into the CA assembly. Under 2 sun illumination, even the lowest temperature at the tip is 30.7 °C is still higher than the environment (Fig. 4g), then the heat only flows out to the environment. Therefore, the evaporation rate of CA assembly above

the theoretical limit originates from the environmental energy input, consistent with the former reports[15,19,20,43,49]. Based on the profiles of temperature distribution and evaporation rates of the CA assembly, we have proposed a model to elucidate the evaporation process. Firstly, for each point on the surface of the conic assembly, the solar thermal input is approximately the same, then it can be derived that the cooler point has the higher evaporation rate owing to more heat inflow from the environment (Supplementary Note 1), i.e., the top evaporates faster than the bottom, thus temperature of the top is lower than that of the bottom, which seems contradictory to the common sense. In this regard, vapor diffusion should be taken into consideration rather than solely temperature. As depicted in Fig. 4i, the holistic evaporation rate should be dominated by two consecutive processes, which are interfacial evaporation and WVD. During evaporation, the water vapor is generated and accumulates near the surface (interfacial evaporation), then diffuses to the surroundings (WVD). Eventually, the system enters into a steady state, where the interfacial evaporation rate $g_{evap}$ equals diffusion flux $g_{w'}$, i.e., $g_{evap} = g_w$. Note that the interfacial evaporation rate ($g_{evap}$) positively correlates with the difference between the saturated pressure ($p^*$) and surface vapor pressure ($p$) ($g_{evap} \propto (p^* - p)$), and the diffusion flux is related to the vapor pressure gradient $\nabla P$ near the surface ($g_w \propto \nabla P$). Combining this interfacial condition with heat conduction and vapor diffusion equations, the corresponding fields of relevant physical quantities could be obtained (finite elemental method is used and details could be found in Supplementary Note 2)

In the following, our calculation results are shown and used to understand the phenomenon that the cooler tip evaporates faster. In Fig. 4h, surface vapor pressure for the bottom ($p_2$) is much higher due to the limited vapor diffusion, resulting in smaller driving force ($\triangle p_2 < \triangle p_1$) for evaporation as well as evaporation rate. In terms of energy flow, the faster evaporation leads to larger energy consumption, resulting in lower surface temperature, which in turn makes thermal energy (heat) flow from the environment to the evaporator ($Q_{in\text{-}envir} > 0$) (Fig. 4g).

As a result, WVD is the rate-determining step, especially for rapid evaporation, which indeed causes the failure of exploiting environmental energy under 2 sun illumination. It is of great significance to promote WVD to further improve evaporation rate. There are two main ways: (i) broadening the effective diffusion space, which has already been achieved by extending evaporators from 2D to 3D[15,17], and (ii) disturbing the near atmosphere by dynamic evaporation, which is much more effective as it directly lowers the water vapor near the surface. In this case, surface vapor pressure $p$ near the surface could be decreased to that of the surroundings ($p_{surr}$), the driving force for evaporation can be boosted to the utmost ($p^* - p_{surr}$).

## Dynamic evaporation of the CA assembly

On account of the understanding above, a dynamic evaporation process was developed based on the reconfigurable $Fe_3O_4@G$ assembly. As shown in Fig. 5a, when applying a variable magnetic field, the orientation of the CA assembly changes to the opposite direction of the magnet, leading to a periodic circular movement (see Supplementary Movie 2). The corresponding optical images and SEM images are displayed in Fig. 5b, c and Supplementary Figs. 30–32, respectively. During rotation movement, both the macroscopic CA shape and the microscopic hierarchical porous structure remained stable, ensuring the enlarged evaporation area and the sufficient water supply for dynamic evaporation. As illustrated in Fig. 5d, the evaporation rate was linearly increased as the rotation rate grew from 0 to 100 rpm under 1 sun illumination, then was greatly improved up to 2.05 kg m$^{-2}$ h$^{-1}$ when the rotation rate was over 100 rpm. The evaporation rates are about 23% higher than that of the static evaporation and theoretical limit under 0.5–2 sun illumination, indicating that the dynamic

evaporation process largely promoted WVD process as well as the evaporation performances.

## Mechanisms underlying the enhancement effect of dynamic evaporation

Benefiting from the dynamic movement of the CA assembly both at macroscopic and microscopic scale, there are three aspects mainly contributing to the evaporation enhancement: (i) the consistency of sunlight absorption at different tilt angle, (ii) the promoted WVD during rotation, and (iii) the improved thermal conductance, water transportation and ionic diffusion resulting from the rapid rearrangement of the nanoparticles. Figure 6a shows that the temperature increasing profiles under 1 sun illumination. CA assemblies at oblique and vertical states present the similarly high temperature, indicating that there is no interference with the sunlight absorption and the solar-thermal effect when changing the orientation of CA assemblies. On a macroscopic view, the dynamic movement not only carries the water vapor away from the evaporation surface but also exposes the CA assembly to the atmosphere with lower vapor pressure. Hence, the lower water vapor pressure near the surface of CA assembly led to the increase of the evaporation rate, which is clearly reflected by the in situ infrared imaging monitor (Fig. 6b and Supplementary Movie 3). Compared with static evaporation, the average temperature of a single CA assembly during dynamic evaporation process at 150 rpm was lowered from 29.2 to 27.8 °C (Fig. 6c). This tendency is consistent with that of the evaporation rate (Fig. 5d). As a result, the dynamic motion greatly promoted the WVD, thus improving the evaporation rate. Besides, the lower slope of the temperature distribution plots also demonstrated that the thermal conductance was getting better as rotation rate increased.

Additionally, Monte Carlo simulation gives a specific picture of the $Fe_3O_4@G$ nanoparticles during dynamic evaporation. The unique shape of CA assembly was endowed by the competition of properly controlled interactions concerning van de Waals forces, magnetic dipoles interactions, gravitational forces, and magnetic forces according to Rosensweig instability[50]. During dynamic rotation, the macroscopic conic shape remains stable, while the arrangement of $Fe_3O_4@G$ nanoparticles is rapidly and completely messed up to a disordered state (Fig. 6d). Meanwhile, it can be seen from the trajectory of four individual nanoparticles (Fig. 6e and Supplementary Movie 4), a single nanoparticle can almost pass by the whole space of the CA assembly, indicating the rapid and thorough rearrangement of $Fe_3O_4@G$ nanoparticles. The speediness of the microscopic rearrangement is also confirmed by the mean square displacement $<r^2>$ derived from the nanoparticles' coordinates statistically (Fig. 6f), which also accounts for the stability of the macroscopic conic shape (Supplementary Note 3). For the sake of clarity, an ideal model of rigid body is employed as contrast, in which nanoparticles don't change their relative positions during rotation through the CA shape is the same. As can be seen, for the dynamic rotation, $<r^2>$ increases much faster than that of the rigid body, and reaches a stable value comparable to the highest level of the rigid body, indicating the rapid rearrangement of the nanoparticles in the CA assembly.

As the nanoparticles move with carrying heat and adjacent liquid, an inner circulation is established which greatly improves the thermal conductivity, water transport, and ionic diffusion. Experimental proofs of elemental and luminescence tracing can also corroborate the simulation. For elemental tracing, pristine $TiO_2$ nanoparticles were laid on the bottom of the CA assembly at the beginning, thus Ti element only existed at the bottom as shown in Fig. 6g i, ii. After dynamic rotation, the Ti element uniformly distributed all over the CA assembly, indicating the complete rearrangement of the nanoparticles (Fig. 6g iii, iv and Supplementary Figs. 39–44). To further confirm the microscopic rearrangement, sodium fluorescein was dripped on the top of the CA assembly to reflect the dynamic process. Under the

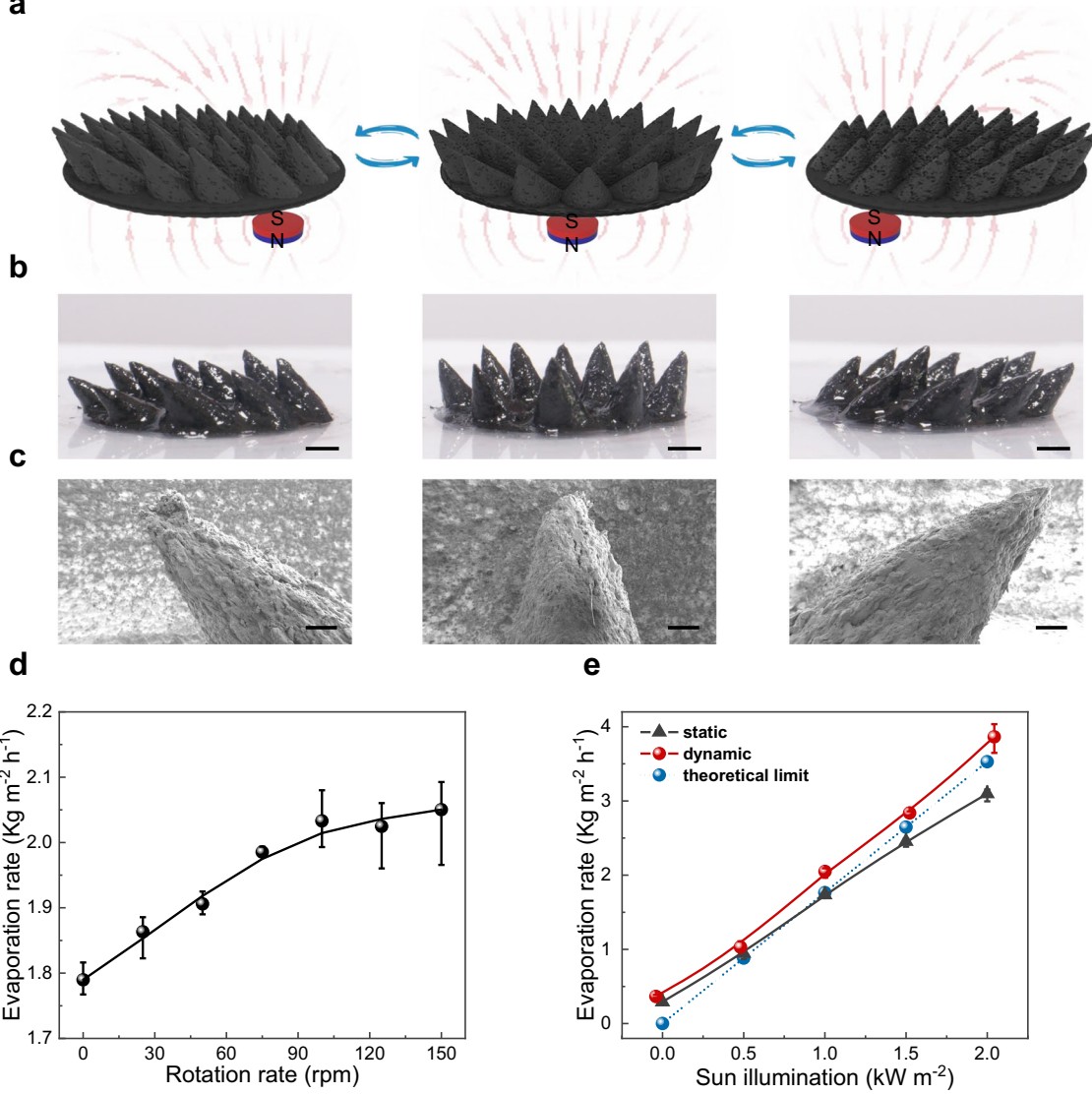

**Fig. 5 | Dynamic evaporation of the CA assembly. a** Schematic diagram depicting the dynamic movement of the conic array (CA) assembly under a variable magnetic field. **b** Real pictures of the CA assembly, from left to right are the left-tilt, vertical, and right-tilt CA assemblies. Scale bar, 2 mm **c** SEM images of an individual CA assembly corresponding to the pictures above. Scale bar, 500 μm **d** Evaporation rate of the CA assembly at dynamic evaporation under 1 sun illumination (n = 3, error bars: standard deviation). **e** Evaporation rate of CA assembly at static and dynamic evaporation under different illumination intensity. The blue dashed line indicates the theoretical limit (n = 3, error bars: standard deviation). Source data are provided as a Source Data file.

irradiation of a 450 nm laser, the position of fluorescein was clearly recognized. As can be seen from Fig. 6h and Supplementary Movie 5, the green fluorescein remained nearly unchanged for static state, while quickly faded within 10 s after dynamic motion because of the diffusion of fluorescein from surface into bulk as well as the water transport from inside out. These results verified the enhanced water transportation and ionic diffusion process resulting from the rapid microscopic rearrangement of nanoparticles.

**Dynamic evaporation of the magnetic hierarchical assembly**

To further promote the WVD process, a magnetic hierarchical assembly was constructed by the properly designed magnetic forces among macroscopic magnets and microscopic $Fe_3O_4@G$ nanoparticles (Fig. 7a). As can be seen from the magnified images shown in Fig. 7b, c, CA assemblies were well configured at the top and the bottom, and the vertical standing stalks repel to each other to enlarge the space for efficient vapor diffusion. As verified by the infrared image (Fig. 7d), it is noteworthy that nearly every part of

magnetic hierarchical assembly owns lower temperature than environment temperature, indicating that the rational hierarchical design promotes WVD, thus leading to more energy gaining from environment and higher evaporation rate. Excitingly, the static evaporation rate under 1 sun illumination reached up to a record high level of 4.80 kg m$^{-2}$ h$^{-1}$. On this basis, the dynamic evaporation could further boost the evaporation rate to a much higher level of up to 5.88 kg m$^{-2}$h$^{-1}$, which is far beyond the theoretical limit (Fig. 7e). Similar to that of CA assembly, even lower surface temperature was found for magnetic hierarchical assembly during dynamic evaporation (Fig. 7d). Meanwhile, a gradually descending tendency was also recorded with rotation rate increasing (Fig. 7f). The temperature decreased rapidly with rotation rate going up from 0 to 75 rpm, and became constant after rotation rate was beyond 75 rpm, corresponding to the evaporation rate increasing trend as shown in Fig. 7g. This consistency further corroborates that the dynamic evaporation induced WVD promotion has greatly endowed to the enhancement of evaporation rate.

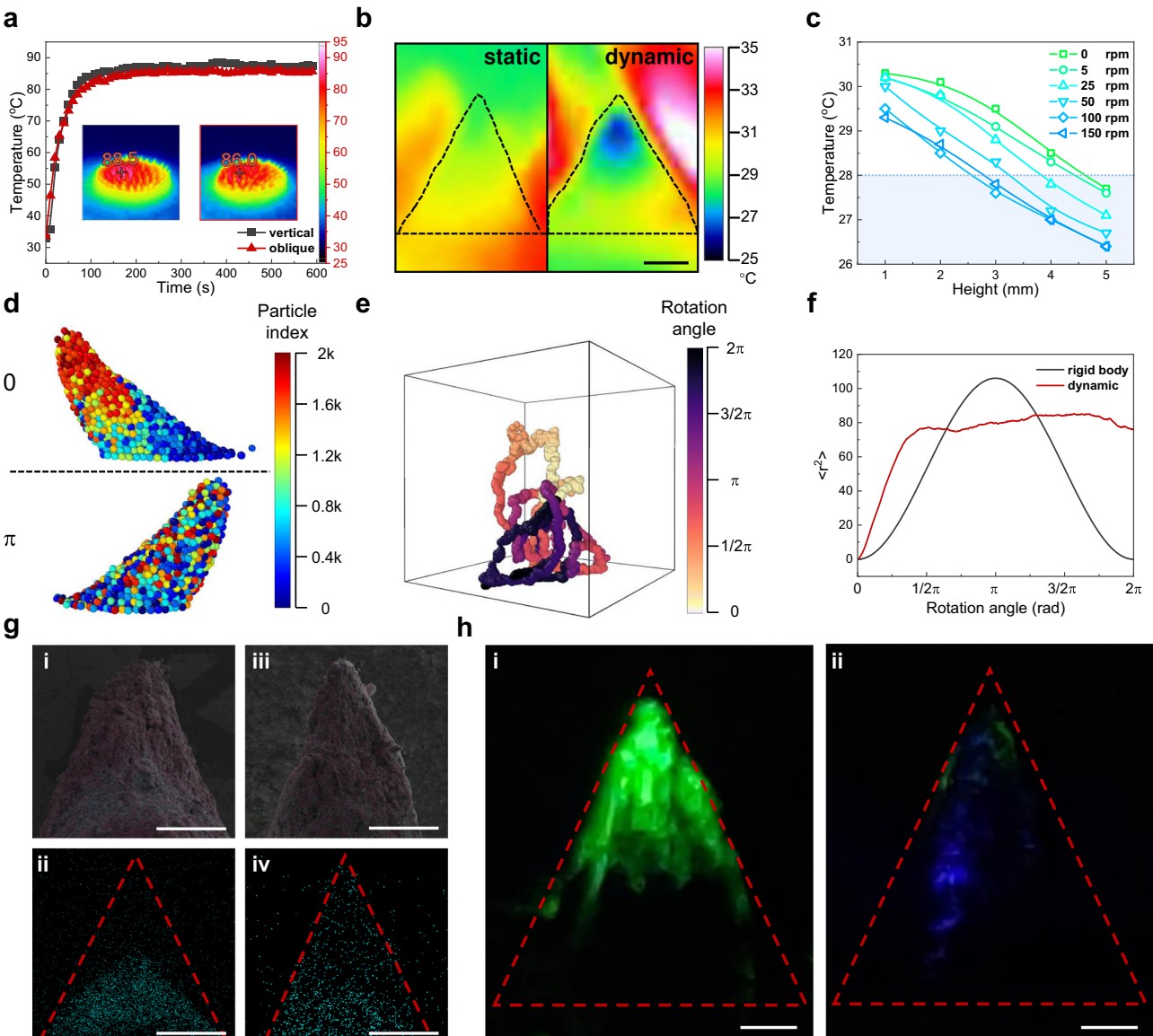

**Fig. 6 | Mechanisms underlying the enhancement effect of dynamic evaporation. a** Temperature variation curves of the vertical and oblique conic array (CA) assemblies upon being radiated under 1 sun illumination. Insets are the corresponding infrared images at thermal equilibrium state. **b** Infrared images of CA assembly at static and dynamic evaporation under 1 sun illumination. Scale bar, 2 mm. **c** Plots of surface temperature of a CA assembly with respect to height under 1 sun illumination at different rotation rate. The blue dashed line is the indication of environment temperature. **d** Pictures of the model resulting from Monte Carlo simulation. The top panel represents the initial state at the beginning of rotation, and the bottom panel depicts the structure after rotation. Particles are colored by the particle index to show the relative position before and after rotation. **e** Trajectory of four adjacent particles during dynamic rotation. The trajectory is colored based on the rotation angle to show the timeline, and the box shows the range of motion of the CA assembly during rotation. **f** Statistical results of the rotation process of rigid body and rearrangement model, the value of $<r^2>$ is the mean square radius of motion of the nanoparticles. **g** Elemental tracing results by using $TiO_2$ nanoparticles as probes to show the particles' relative movement, **i** and **ii** are the composite image and Ti element distribution image at the beginning of rotation, while **iii** and **iv** correspond to the results after dynamic rotation. Scale bar, 2 mm. **h** Luminescence tracing results of the dynamic evaporation to show the ions movement, green fluorescence is observed at the top at the beginning (**i**), and quickly fades out after dynamic rotation within 10 s (**ii**). Scale bar, 1 mm. Source data are provided as a Source Data file.

## Discussion

We developed a reconfigurable and magnetically responsive solar evaporator taking advantage of the reversible assembly of $Fe_3O_4@G$ nanoparticles. The unique conic array (CA) assembly induced by magnetic field endows the evaporator with excellent characteristics including good salt resistant ability, adequate water transport, recycling ability as well as high evaporation rate. The dynamic reconfiguration of CA assembly both at macro and microscopic scale leads to a more than 20% enhancement in evaporation rate compared with traditional static evaporation. Further, a sophisticated hierarchical assembly was constructed through the rational combination between macro magnets with CA assemblies, and a record high evaporation rate of up to 4.80 kg m$^{-2}$ h$^{-1}$ was achieved under 1 sun illumination. With dynamic evaporation, even a much higher evaporation rate of up to 5.88 kg m$^{-2}$ h$^{-1}$ was realized. Consequently, the dynamic and reconfigurable assembly enabled by reconfigurable $Fe_3O_4@G$ nanoparticles has provided a new perspective to further improve water evaporation performance, which could contribute a basic understanding into the field of ISSG, as well as the rational design of solar water evaporation systems in the future.

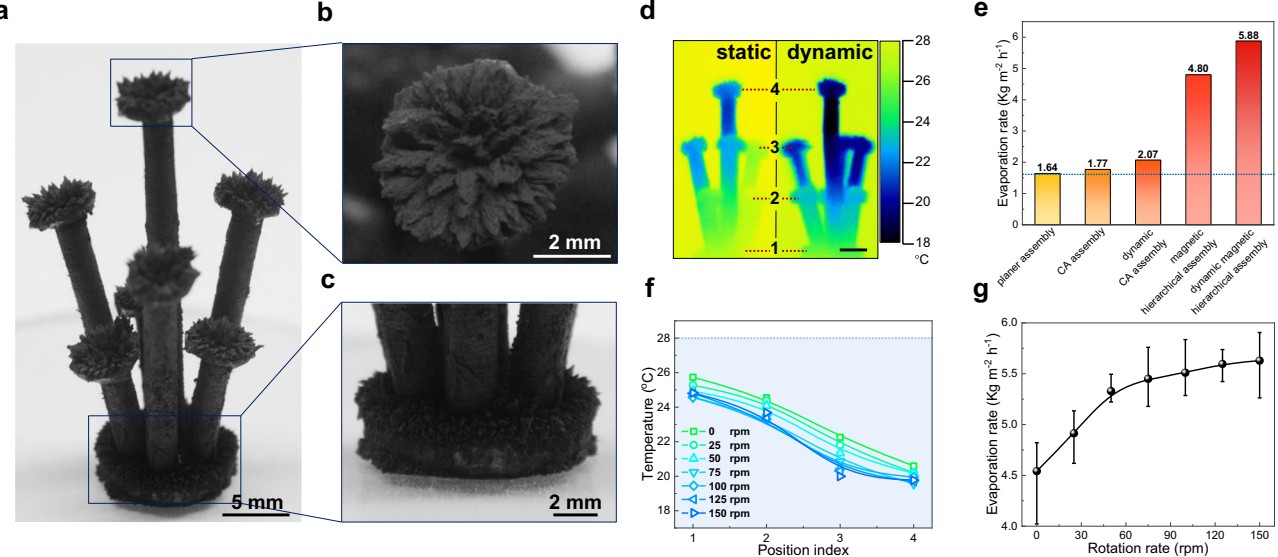

**Fig. 7 | Dynamic evaporation of the magnetic hierarchical assembly.**
**a**–**c** Pictures of the magnetic hierarchical assembly. A full-body photograph (**a**), the conic array (CA) assemblies constructed on the top (**b**) and at the bottom (**c**). **d** Infrared images of magnetic hierarchical assembly at static and dynamic evaporation under 1 sun illumination. Scale bar, 2 mm. Numbers 1–4 stands for the specified position in the picture. Scale bar, 5 mm. **e** Histogram of evaporation rates of planar assembly, CA assembly, dynamic CA assembly, magnetic hierarchical assembly, and dynamic magnetic hierarchical assembly under 1 sun illumination. **f** Plots of surface temperature of magnetic hierarchical assembly with respect to height under 1 sun illumination at different rotation rate. The blue dashed line is the indication of environment temperature. The position index corresponds to the specified position in (**d**). **g** Evaporation rate of the magnetic hierarchical assembly at different rotation rates under 1 sun illumination (*n* = 3, error bars: standard deviation). Source data are provided as a Source Data file.

## Methods

### Preparation of the GO dispersion

GO dispersion were synthesized through the modified Hummers' method[51] from natural graphite powder (325 mesh, Qingdao Huatai lubricant sealing S&T Co., Ltd.). Typically, 5 g graphite powder was mixed with 115 mL $H_2SO_4$ (98 wt%) at 0 °C under mechanically stirring. Afterwards, 15 g $KMnO_4$ was slowly added to the above mixed solution followed by the addition of 200 mL deionized water. Then, the mixture was transferred into a 95 °C oil bath and kept reaction for another 30 min. Next, the mixture was poured in 1500 mL deionized water and 30 wt% $H_2O_2$ solution was added subsequently to quench the reaction until no bubbles emerged. Prior to dialysis (cut off: 8000–14,000 Da), the above mixture was filtered and washed with HCl solution (1 mol L$^{-1}$) for three times. Finally, the purified GO suspension was obtained by centrifugation at 2000 rpm and 8000 rpm. For the consequent synthetic steps, GO sheets were torn apart into small-sized GO sheets by ultrasonication for 30 min using a cell pulverizer.

### Preparation of the Fe₃O₄@G nanoparticles

Firstly, 5 g $Fe_3O_4$ nanoparticles (Alab Chemical Technology Co. Ltd) were dispersed in 500 ml deionized water, followed by the addition of 50 ml 2 mg mL$^{-1}$ GO dispersion. Afterwards, the mixture was ultrasonicated for 20 min by ultrasonic cleaning machine to thoroughly disperse $Fe_3O_4$ nanoparticles in the solution, and subsequently was vigorously stirred at 400 rpm overnight to let nanoparticles be combined with GO sheets. Thereafter, the $Fe_3O_4$@GO nanoparticles were collected by magnetic decantation and washed with deionized water for at least three times, then were dried via vacuum drying method at 60 °C for 48 h. Finally, $Fe_3O_4$@G nanoparticles were gained through reduction of the $Fe_3O_4$@GO nanoparticles by high temperature calcination at 400 °C under $N_2$ atmosphere for 1 h.

### Configuration and reconfiguration of the Fe₃O₄@G assemblies

Taking CA assembly as an example, as shown in Supplementary Fig. 13, $Fe_3O_4$@G nanoparticles were firstly mixed with water to become a slurry. A round-shaped NbFeB magnet was placed beneath the slurry in a petri dish, different interval distances were applied to control the magnetic field intensity. Rotation of the magnet was taken to give a disturbance on the $Fe_3O_4$@G slurry and an ensuing conic-shaped pattern structure could be obtained. Thereafter, the free-standing CA assembly could be obtained via the evaporation-induced self-assembly. For the reconfiguration, the disassembly process can be simply accomplished by water rinsing with vibration. Consequently, the $Fe_3O_4$@G nanoparticles were collected easily by magnetic force, and thus ready for the next configuration.

### Measurement of the static solar water evaporation performance

For the construction of static evaporation apparatus, as shown in Supplementary Fig. 14a glass fiber membrane was used as the substrate of the evaporator, beneath was a thermal insulator made of polystyrene foam, and cotton swab was fixed to reach the water surface as the water path. Thereafter, the evaporation apparatus was fixed on the analytical balance, which is connected with a computer to record the real-time mass change. For the water to be evaporated, deionized water and NaCl solutions with different concentration were adopted as freshwater and salt water respectively. The temperature and humidity of the room was controlled at about 28 °C and 50% respectively. Besides, wind and other disturbances were carefully avoided from the evaporator during the test.

### Measurement of the dynamic solar water evaporation performance

Apart from the dynamic evaporation apparatus, other testing conditions are the same with static evaporation test. As for the dynamic evaporation apparatus, a holder was designed to support the evaporator. The glass fiber membranes are used as the water buffer reservoir. For the long term evaporation, more water can be replenished continuously by a peristaltic pump through the water pipe. Below the holder and the shelf is a magnet stuck on a rotation tray (see pictures in Supplementary Fig. 28). The motor was connected to the rotation tray, and different rotation rate could be controlled. When the magnet moves, the magnetic field near the

evaporator will change, leading to the reconfiguration of the CA assembly. As for the dynamic evaporation for the magnetic hierarchical assembly, the holder is abandoned. The glass fiber membrane at the bottom is the water buffer reservoir, water can be replenished continuously by a peristaltic pump (Supplementary Fig. 46).

### Element tracing of the nanoparticles' movement inside the CA assembly

$TiO_2$ nanoparticles with particle size around 100 nm (Shanghai Macklin Biochemical Co., Ltd.) were used as probe because it does not respond to magnetic field thus should not move by itself in the CA assembly. At the beginning, $TiO_2$ nanoparticles were mixed with $Fe_3O_4$@G nanoparticles by mass ratio of 1:1 to be a mixed slurry. Then the mixture was carefully laid on the top or at the bottom of the CA assembly. Consequently, the CA assembly underwent a dynamic rotation process at rotation rate of 150 rpm, followed by natural drying to become a freestanding CA assembly for EDS mapping (Gemini300, Zeiss).

### Luminescence tracing of ions movement inside the CA assembly

Fluorescein sodium (Saan Chemical Technology Co., Ltd.) was used as the fluorescence tracer, and a laser pen with 450 nm emission wavelength was used as the laser source. At the beginning, the CA assembly was well configured on the apparatus, then the fluorescein sodium solution (1 M, 2 μL) was carefully dripped on the top of the conic assembly. The fluorescence was recorded by a camera, and the rotation rate of 150 rpm was used for dynamic process.

### Monte Carlo simulations

To track the nanoparticles' movement inside the CA assembly during dynamic evaporation, Monte Carlo simulations[52] were carried out and the calculation model is shown in Supplementary Fig. 37a, where 2000 individual nanoparticles were spontaneously assembled induced by the competition of the magnetic forces, magnetic dipole interactions, van de Waals interactions, and gravity forces. The initial configuration was gained after relaxation to an equilibrium state in a static sloping magnetic field. Afterwards, the magnetic field was controlled to rotate, along with the rotation of the oblique conic assembly.

In detail, the related potential function includes van de Waals interaction, magnetic dipole interaction, magnetic force, and gravity force, which are listed below:

Van de Waals interaction is represented by Lennard-Jones potential,

$$E_{vdw} = \sum_{i>j} 4\varepsilon \left[ \left( \frac{\sigma}{r_{ij}} \right)^{12} - \left( \frac{\sigma}{r_{ij}} \right)^{6} \right],$$ (1)

where $\varepsilon$ and $\sigma$ are the two parameters of Lennard-Jones potential, for nanoparticles $\varepsilon = 0.003$, $\sigma = 1$, $r_{ij}$ represents the distance between particle $i$ and particle $j$.

Magnetic dipole interaction is given by

$$E_{md} = \sum_{i>j} \frac{\mu_0}{4\pi r_{ij}^3} \left[ \boldsymbol{M_i} \cdot \boldsymbol{M_j} - 3\left(\boldsymbol{M_i} \cdot \hat{\boldsymbol{r}}_{ij}\right)\left(\boldsymbol{M_i} \cdot \hat{\boldsymbol{r}}_{ij}\right) \right],$$ (2)

where $\mu_0$ is the permeability of vacuum which is $4\pi \times 10^{-7}$ N A$^{-2}$, $\boldsymbol{r}_{ij}$ is the distance between particle $i$ and particle $j$, $\boldsymbol{M_i}$ and $\boldsymbol{M_j}$ are the magnetic moment, $\hat{\boldsymbol{r}}_{ij}$ is the unit vector from particle $i$ to particle $j$. In our calculation $\mu_0/4\pi r_{ij}^3$ is set to 0.03, and the magnitude of $\boldsymbol{M_i}$ is restricted to no more than 1.

The potential energy in magnetic field is given by

$$E_m = \sum_i \boldsymbol{B} \cdot \boldsymbol{M_i},$$ (3)

where $\boldsymbol{B}$ is the magnetic flux density, $B_z = 2$ and $B_x^2 + B_y^2 = 1$ are set to simulate the magnetic rotation.

The gravitational potential energy is represented by

$$E_g = \sum_i mgh_i,$$ (4)

where $m$ is the mass of a particle, $g$ is the gravitational acceleration, $h_i$ is the height of particle $i$. mg together is set to 0.15 in our calculation.

The Monte Carlo simulations are performed with thermal energy, i.e., $K_B T$, equals 0.0345. 640,000 and 160,000 Monte Carlo steps are used for simulating one circle of rotation to figure out the influence of rotation velocity. More details about the program can be provided in Supplementary Note 4 and Supplementary Fig. 35.

Note that the ratio between different energies set in our calculations generally follows the experiments. To be specific, the typical energies in experiments are the parallel dipole–dipole interaction energy $3.29 \times 10^{-18}$ J, the largest gravity potential energy $2.54 \times 10^{-16}$ J, maximum energy of the magnetic dipole under external magnetic field $7.26 \times 10^{-17}$ J (see Supplementary Table 2).

## Data availability

The data generated in this study are provided in the Supplementary Information and Source Data file. Source data are provided with this paper.

## Code availability

The code supporting this study is available from the corresponding author upon request.

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

## Acknowledgements

This work is supported by the financial support from the National Science Foundation of China (No. 52073159, 52022051, 22035005, 22075165, 52090032), State Key Laboratory of Tribology (SKLT2021B03), and Tsinghua-Foshan Innovation Special Fund (2021THFS0501). F.L. also acknowledges the support from the National Science Foundation of China (11972349, 11790292) and Strategic Priority Research Program of the Chinese Academy of Sciences (Grant No. XDB22040503).

## Author contributions

L.Q. and Y.H. designed the experiments. Y.H. performed the majority of experimental measurements with help from M.W., T.L., H.M., H.Y., and H.C. Y.H., and F. L. conducted theoretical simulation. Y.H. prepared the manuscript under the advice given from L.Q., H.C., and F.L. All authors discussed the results and reviewed the manuscript. L.Q. supervised the entire project.

## Competing interests

The authors declare no competing interests.
