## [Peer Review File · Nature Communications]

A reconfigurable and magnetically responsive assembly for dynamic solar steam generationREVIEWER COMMENTS

Reviewer #1 (Remarks to the Author):

The present manuscript described an exciting and original work dealing with the use of novel, dynamically configurable, magnetic structures to enhance the interfacial evaporation of water. The evaporation occurs at a surface composed of graphene encapsulated Fe₂O₃ nanoparticles, which can form conical structures under applied magnetic fields. These arrays are reconfigurable because the independent particles can respond very rapidly to a changing magnetic field, reshaping the conical structure to different inclinations. In the process, the particles shuffle, enhancing the heat and mass transfer across the system and, therefore, water evaporation. With the more complex structures tested, the enhancement in evaporation is considerable compared to a flat array, which still improves with the application of a rotating magnetic field. The results are original and relevant, and from my point of view, the proposed structures could have a significant technological impact in the future. Also, in my opinion, the authors have carried out a thorough work, and the information provided is quite complete and convincing, clearly supporting their main conclusions. Therefore, I consider this material valuable and appropriate for publication in Nature Communications, with only a few minor observations:

- 1) In lines 229-231, it is unclear how a one-dimensional heat conduction model was applied to the conical assembly; what does the x coordinate represent? Please explain more carefully.
- 2) The "optional diffusion direction angle" concept is not very clear; please explain or give a reference.
- 3) A careful check of English should be carried out to correct grammar problems. For instance: in line 49, "... in near circumstances around evaporator ..."; line 352, "...thus Ti element only distributed at the bottom..."; etc.

Reviewer #2 (Remarks to the Author):

The authors are well aware of the current problems in solar steam generation, and presented creative solutions using dynamic reconfiguration and reassembly. Through magnetically responsive behavior of the conic arrays using the controllable and reversible assembly of Fe₃O₄@G, they have newly solved the insufficient water vapor diffusion problems in the static evaporation systems. I think this research is a truly influential and substantial achievement that can accelerate the practical use of solar-steam desalination. Therefore, I think the paper could be published in NC after these concerns are addressed. Below are the specific comments.

1. The introduction needs to be improved. Particularly more details need to be added. The literature review failed in covering the state-of-art of studies in the solar steam field, which is critical for readers to understand the research gap this study aimed to address.
2. As shown in Figure 4a, the static evaporation can absorb water via the water path. How to maintain the specific shape of the evaporator during the long-term evaporation? If there is magnetic force, what's the position relationship among the evaporator, magnet, and the water reservoir? Besides, in the dynamic evaporation, where is the water reservoir (Supplementary Figure 25)?
3. Line 293-295, the author pointed that "the evaporation rate was linearly increased as the rotation rate grew from 0 rpm to 100 rpm under 1 sun illumination", but why does the evaporation rate increase slowly at the higher rotation rate (Figure 5d)?
4. The dynamic evaporation rates are about 23% higher than that of the static evaporation, how about the performance with a convective flow?
5. As shown in Figure 6a and 6b, why the highest temperature is different under 1 sun illumination?
6. As shown in Figure 7a, how to construct the magnetic hierarchical assembly, especially the joint between the stalk and the flower? In its dynamic process, what's the effect of magnetic field on the stalk? What's the desalination effect and stability for the magnetic hierarchical assembly?
7. In Figure 7, the magnetic hierarchical assembly leads more energy gaining from environment. The energy consumption in the system should be calculated. The explanations should be given to ensure the result. see the literatures Adv. Funct. Mater. 2019, 29, 1903255; Joule, 2018, 2, 1331-1338.

8. The evaporation enthalpies of water in Fe₃O₄ and Fe₃O₄@GO are lower than that of bulk water, but that of water in Fe₃O₄@G is higher, in Supplementary Figure 19, the author should give some explanations about the change in the evaporation enthalpies.

9. Line 207-211, "Under illumination of 0 – 1000 kW m⁻², the evaporation rate of CA assembly is higher than the theoretical limit, which is calculated from the evaporation enthalpy^{12,13,41} (Supplementary Fig. 19). However, it becomes lower than the theoretical limit when illumination larger than 1000 kW m⁻²." The author should check the expression of "1000 kW m⁻²", the highest intensity in the text is only 2 sun illumination.

Reviewer #3 (Remarks to the Author):

This work is devoted to an important area related to the development of new efficient devices for obtaining fresh and clean water. The method proposed by the authors refers to solar desalinators. Clean water can be obtained by evaporation of seawater or wastewater with its subsequent condensation. The use of solar energy is a budget-friendly and environmentally friendly approach, but, as a rule, inefficient. For this reason, the authors are investigating the possibility of increasing the evaporation rate by using a ferromagnetic liquid. A colloidal solution consisting of iron oxide nanoparticles and water is applied to the surface of the device. Graphene has been added to ensure the stability of the system, which prevents nanoparticles from agglomeration. Due to the influence of the magnetic field, conical structures and other forms of free surface are created. The black color of the solution contributes to a good absorption of solar radiation. In addition, a non-flat surface increases the evaporation area. An additional increase in the evaporation rate is given by an alternating magnetic field, which causes the dynamics of cone assemblies and other structures. As the water evaporates from the free surface of the structure, liquid flows from the container along a special path. The authors investigate the evaporation rate in a static and dynamic structure, how the rate of change of the magnetic field affects this process, the influence of the shape of the structure on the evaporation rate and in which areas evaporation is most intense. The authors also note that in this approach there is practically no crystallization of salt on the free surface, which is an additional advantage. A series of full-scale experiments and numerical calculations have been carried out in this work, which make it possible to deduce the main effects and mechanisms. The text of the manuscript is written in a clear language. Of course, this work is important for the further development of this area. The lack of clean water is one of the world's biggest problems, especially for the countries of the African continent. But, nevertheless, before publishing this manuscript, it is necessary to deal with a number of problematic places, which I will mention below.

Lines 58-61. Here you need to further explain. Due to what is the transportation of water? Filtration of liquid due to the capillary effect in a porous medium formed by voids between nanoparticles in a cluster?

Lines 232-234. In my opinion, the tip of the conical structure is relatively colder, because evaporation is more intense there, since the gradient of vapor concentration near the surface is higher in that area. After all, evaporation also occurs near the cone from the horizontal surface. To describe this mathematically, it is necessary to solve numerically not only the heat equation for the liquid phase, but also the equation of vapor diffusion in the air region. In addition, the relatively rapid accumulation and crystallization of salt in the area of the tip of the conical structure additionally indicates that evaporation occurs more intensively in that area, respectively, the capillary flow transfers more dissolved matter to this area. What do you think about this?

Lines 248-249. Based on equation (1), it is difficult for me to understand why the cold tip evaporates faster. I have already expressed my opinion on this matter above. How does the theta angle affect the evaporation rate? In the Supplementary Information, eq. (4) is referred to as Fick's first law. In Fick's first law, there must be a derivative in space that is not in (4). How does the theta angle appear in (4)?

Lines 260-262. "ii) disturbing the near atmosphere by dynamic evaporation, which is much more effective as it directly eliminates the water vapor gradient near the surface." Eliminate? Maybe, on the contrary, you need to increase the vapor gradient at the surface? After all, the higher the

gradient, the more intense the evaporation occurs. I think you should to remove the word "gradient".

Does this mean adding an air flow to the system in order to remove vapor even faster from the surface of the evaporator, thereby speeding up the process of liquid-vapor transition?

Fig. 4i. I believe that in this schematic Fig. 4i at the bottom there are not enough arrows indicating the evaporation flux density directed from the horizontal free surface on both sides of the cone. This is the key point in explaining that the vapor concentration gradient will be higher near the tip of the cone (see my comment for Lines 232-234).

Lines 339-340. "mean square radius of motion". Does this mean "mean squared displacement" (MSD)? It's a common term.

Line 511. "2000 individual nanoparticles". With such a number of particles, the question immediately arises. Does the size of the cone structure of the model correspond to the size of the same structure from the experiment? What is the approximate size of nanoparticles? If the dimensions of the structures differ many times, then this point in the model needs to be further justified. It should be explained why, with a significant difference in scale, the model will describe the experiment.

Lines 512-513. Why does the model not take into account the presence of liquid in a porous medium and capillary flow? A mixture of particles and liquid is a solution. Doesn't the viscosity of the solution have an effect on this process? In the absence of a liquid, won't particles go away from this structure when exposed to a magnetic field?

Line 536. A table should be added indicating all the parameters of the problem, their values and units of measurement.

Lines 537-539. There is no description of the calculation algorithm here. It is necessary to provide at least a brief description or in the form of a pseudocode. Only formulas of particle interaction energies are indicated. How was the particle dynamics simulated and the particle collision processed? What is the essence of the Monte Carlo method used here? For what quantities was a random number generator used? Which random number generator was involved? Has an existing package been used or has a computer program been developed?

Reviewer #1 (Remarks to the Author):

The present manuscript described an exciting and original work dealing with the use of novel, dynamically configurable, magnetic structures to enhance the interfacial evaporation of water. The evaporation occurs at a surface composed of graphene encapsulated Fe₂O₃ nanoparticles, which can form conical structures under applied magnetic fields. These arrays are reconfigurable because the independent particles can respond very rapidly to a changing magnetic field, reshaping the conical structure to different inclinations. In the process, the particles shuffle, enhancing the heat and mass transfer across the system and, therefore, water evaporation. With the more complex structures tested, the enhancement in evaporation is considerable compared to a flat array, which still improves with the application of a rotating magnetic field. The results are original and relevant, and from my point of view, the proposed structures could have a significant technological impact in the future.

Also, in my opinion, the authors have carried out a thorough work, and the information provided is quite complete and convincing, clearly supporting their main conclusions. Therefore, I consider this material valuable and appropriate for publication in Nature Communications, with only a few minor observations:

Many thanks for your valuable comments and good points. According to your remarkable questions and kind advice, we have updated our theoretical mechanism model to a complete form and numerically solved it by the finite element method in the revised manuscript. The following are the point-to-point replies to the comments.

1. In lines 229-231, it is unclear how a one-dimensional heat conduction model was applied to the conical assembly; what does the x coordinate represent? Please explain more carefully.

Reply:

It is a simplified model that may cause confusion. We have substituted it with numerically solved thermal conduction differential equations. More specific diagrams and information are given in the revised manuscript. Please check Fig 4. g-i in the revised manuscript and Supplementary Note 2 in supplementary information for details.

Below is the explanation for the one-dimensional heat conduction model.

As the CA assembly has a conic shape, it has $C_{\infty v}$ symmetry, then one of its cross-hatching profiles could stand for all the rest, which is what Fig 4i depicted. At the same time, this profile is bilateral symmetrical, thus we can only investigate the right side.

For thermal steady state, on the bevel of the conic assembly, the heat source of sunlight should be equal to the heat consumption of evaporation plus the heat conduction with the environment. Then the thermal conduction model is simplified to a one-dimensional problem (the hypotenuse on the right). Here, x represents the position coordinate of the point on the line. The thermal conduction equation can be given by $d^2T/dx^2 = 0$, which indicates the linear correlation between the height (corresponding to x) and the surface temperature.

2. The “optional diffusion direction angle” concept is not very clear; please explain or give a reference.

Reply:

This concept is also a parameter proposed by us to simplify the equations, aiming to indicate the influence of the space narrowness on the diffusion flux. To clarify this, complete equations for this system have been provided in the revised supplementary information, and are numerically solved by the finite element method. Graphical diagrams are clear and accurate to describe the physical image, and the “optional diffusion direction angle” is abandoned to avoid the possible misleading information coming from the simplification of the equations. Please check Fig 4. g-i in the revised manuscript and Supplementary Note 2 in supplementary information for details.

3. A careful check of English should be carried out to correct grammar problems. For instance: in line 49, “... in near circumstances around evaporator ...”; line 352, “...thus Ti element only distributed at the bottom...”; etc.

Reply:

We have double-checked the grammar and made corrections in the revised manuscript.

Reviewer #2 (Remarks to the Author):

The authors are well aware of the current problems in solar steam generation, and presented creative solutions using dynamic reconfiguration and reassembly. Through magnetically responsive behavior of the conic arrays using the controllable and reversible assembly of $\text{Fe}_3\text{O}_4@\text{G}$, they have newly solved the insufficient water vapor

diffusion problems in the static evaporation systems. I think this research is a truly influential and substantial achievement that can accelerate the practical use of solar-steam desalination. Therefore, I think the paper could be published in NC after these concerns are addressed. Below are the specific comments.

Thanks a lot for your positive remarks and constructive suggestions. Based on your questions, additional experiments are conducted to improve the manuscript. Below are the point-to-point replies to your remarkable comments.

1. The introduction needs to be improved. Particularly more details need to be added. The literature review failed in covering the state-of-art of studies in the solar steam field, which is critical for readers to understand the research gap this study aimed to address.

Reply:

In the revised manuscript, the two papers you mentioned are cited. Besides, we have added some state-of-the-art research works (*Adv. Funct. Mater.* **2019** *30*, 1907234; *Energy Environ. Sci.* **2020** *13*, 3646-3655; *Joule.* **2020** *4*, 2733-2745; *Nat. Commun.* **2018** *9*, 5086; *Nat. Commun.* **2022** *13*, 849; *Adv. Funct. Mater.* **2021**, *31*, 2101036). At the same time, as for the introduction part, more detailed analysis of the background and the research gap is given in the revised manuscript.

2. As shown in Figure 4a, the static evaporation can absorb water via the water path. How to maintain the specific shape of the evaporator during the long-term evaporation? If there is magnetic force, what's the position relationship among the evaporator,

magnet, and the water reservoir? Besides, in the dynamic evaporation, where is the water reservoir (Supplementary Figure 25)?

Reply:

1) The CA assembly is a free-standing structure when undergoing static evaporation. To explain this explicitly, we have added a schematic image in the revised supplementary information. As shown in Fig. R1, the conic array can be molded by the external magnetic field. After evaporation induced assembly, a free-standing CA assembly could be obtained. Note that, at this stage, the CA assembly can afford itself during static evaporation. When regeneration is needed, the disassembly process can be accomplished by water rinsing and vibration. Afterwards, disassembled $\text{Fe}_3\text{O}_4@G$ nanoparticles can be involved into the next circulation.

Figure R1. Schematic image of the reconfiguration cycle of $\text{Fe}_3\text{O}_4@G$ nanoparticles.

a Discrete $\text{Fe}_3\text{O}_4@G$ nanoparticles. **b** The magnetic assembly process. **c** The free-standing CA assembly.

2) During static evaporation, no magnet is needed.

3) As shown in Fig. R2, in the case of dynamic evaporation, a small holder is designed to support CA assembly during dynamic evaporation. The glass fiber membranes are used to be the water reservoir. For the long term evaporation, more water can be replenished continuously by a peristaltic pump through the water pipe (Fig. R2c).

Figure R2. Schematic and optical images of the dynamic evaporation apparatus. **a, b** The schematic image (a) and optical image (b) of the dynamic apparatus. **c, d** The schematic image (c) and the optical image (d) of the dynamic holder.

3. Line 293-295, the author pointed that “the evaporation rate was linearly increased as the rotation rate grew from 0 rpm to 100 rpm under 1 sun illumination”, but why does the evaporation rate increase slowly at the higher rotation rate (Figure 5d)?

Reply: It is a good question. The dynamic rotation can promote the vapor diffusion process, the enhancement effect is relatively small at low rotation rate, and becomes higher as rotation rate increases. However, restricted by the water vapor diffusion constant, the atmospheric pressure and humidity of surroundings, there is an upper limit

of the enhancement effect of dynamic rotation, which leads to the plateau at high rotation rate.

4. The dynamic evaporation rates are about 23% higher than that of the static evaporation, how about the performance with a convective flow?

Reply: Additional experiments are conducted to show the influence of the convective flow on the evaporation performance of the CA assembly. Briefly, the dynamic evaporation process and convective flow have the similar effect of promoting WVD. But the dynamic evaporation process can enhance water transportation and prevent salt precipitation, which are unique features that convective flow doesn't possess. Below are the details of the experiments.

As shown in Fig. R3, in the static evaporation process, the evaporation rate increases with the intensity of the convective flow. By comparing with dynamic evaporation, the enhancement effect of the dynamic evaporation is equivalent to that of 0.7 m s^{-1} convective flow. At the moment, the rotation rate of the fan is 950 rpm, which is far larger than that of the dynamic apparatus (150 rpm).

Besides, the tests of the dynamic evaporation under convective flow are also conducted. From 0 m s^{-1} to 0.5 m s^{-1} , the evaporation rate remains unchanged, indicating that the dynamic evaporation and the convective flow has the similar influence on WVD. But for the convective flow larger than 0.5 m s^{-1} , the evaporation rates are higher than that of the static evaporation. This is the contribution of the enhanced water supply induced by the inner circulation of the dynamic evaporation process.

Figure R3. Evaporation rate of the CA assembly under convective flow under 1 sun illumination, and 150 rpm rotation rate for dynamic evaporation.

5. As shown in Figure 6a and 6b, why the highest temperature is different under 1 sun illumination?

Reply: Thanks for your question. In Fig. 6a, the temperatures are recorded when CA assemblies are at dry state (without water evaporation), while in Figure 6b, the CA assembly are undergoing static/dynamic evaporation process, which makes the surfaces temperature lower than that of the dry state. As for the reason why the dry state temperatures are used to draw comparison in Figure 6a, it is because here an investigation is made for whether or not oblique CA assembly could cause sunlight absorption loss. Therefore, water evaporation should be avoided to let the surface temperature to be the sole indicator of solar absorption.

6. As shown in Figure 7a, how to construct the magnetic hierarchical assembly,

especially the joint between the stalk and the flower? In its dynamic process, what's the effect of magnetic field on the stalk? What's the desalination effect and stability for the magnetic hierarchical assembly?

Reply:

1) Basically, the magnetic hierarchical assembly is firstly constructed by magnetic interactions between macroscopic magnets, which can render a corresponding magnetic field on demand. Thereafter, $\text{Fe}_3\text{O}_4@\text{G}$ nanoparticles can be assembled into CA assembly at particular positions according to the magnetic field.

As shown in Fig. R4a-c, the magnetic hierarchical assembly is made of 7 separate small discal magnets (diameter = 5 mm, thickness = 0.5 mm), 10 rod-like magnets (diameter = 2 mm, height = 10 mm), one big discal magnet (diameter = 10 mm, thickness = 2 mm) and $\text{Fe}_3\text{O}_4@\text{G}$ nanoparticles. The N/S poles of the magnets are indicated by the blue/red color in Fig. R4a and d. The flower refers to the small discal magnet with CA assembly on it, and the stalk is the rod-like magnet. They are connected with each other by the attraction of the magnetic force (N pole to the S pole). Besides, the stalks repel each other at the top due to the same polarity, and are attached to the bottom due to a stronger S pole, thus leading to a large-evaporation-area macroscopic figure. Before combined with $\text{Fe}_3\text{O}_4@\text{G}$ nanoparticles, to ensure adequate water supply, a layer of air-laid paper is stuck on the surface of the magnets (Fig. R4e). Eventually, $\text{Fe}_3\text{O}_4@\text{G}$ nanoparticles are attracted on the magnets and spontaneously form CA assemblies on the top and the bottom.

Figure R4. Schematic image of the construction of the magnetic hierarchical assembly.

a-c The schematic and optical images of the macroscopic magnets assembly and the component magnets. **d** The schematic image of the magnetic field of the center connection structure, the ligand notes indicate the magnetic field intensity. **e** The optical images of the air-laid paper wrapped macroscopic magnets. **f** The magnified optical image of the stalk showing the distribution state of the $\text{Fe}_3\text{O}_4@\text{G}$ nanoparticles.

2) As shown in Fig. R4d, the schematic magnetic field lines are depicted, and the magnetic field intensity is measured. According to the Rosensweig Instability, the conical arrays only appear in a vertical magnetic field, hence the CA assembly can only form at the top and at the bottom. At the stalk, abundant $\text{Fe}_3\text{O}_4@\text{G}$ nanoparticles can be attracted on it, while no conic array is formed (Fig. R4f).

3) About the desalination effect of the magnetic hierarchical assembly: Simulated salt

water with salinity varying from 3.5 wt% to 20 wt% is used to test the evaporation rate of the magnetic hierarchical assembly. As shown in Fig. R5, although evaporation rate decreases at high salinity which is caused by the vapor pressure lowering effect, the magnetic hierarchical assembly shows good desalination effect ($> 4 \text{ Kg m}^{-2} \text{ h}^{-1}$) even with 20 wt% saline.

Figure R5. Evaporation rates of the magnetic hierarchical assembly with different salinity under 1 sun illumination.

About the stability of the magnetic hierarchical assembly: For the structural stability, a magnetic hierarchical assembly was tested and laid still for half a year (Fig. R6). It can be seen that the wholistic figure keeps stable (Fig. R6a), and the CA assemblies on the top and at the bottom are still perfect and clear, indicating the robustness and durability of the magnetic hierarchical assembly.

Figure R6. a-e Optical images of the magnetic hierarchical assembly after half a year. The full-length photo (a), CA assemblies at the top (b-d), and the CA assembly at the bottom (e).

Figure R7. The desalination stability of the magnetic hierarchical assembly. a 12 hours' desalination test with 3.5 wt% saline under 1 sun illumination. b Evaporation rates of

the magnetic hierarchical assembly under different circumstances. **c, d** Optical images of the slat contaminated (**c**) and recovered (**d**) magnetic hierarchical assembly.

For the desalination stability, even after half a year, the magnetic hierarchical assembly maintains fast evaporation rate of up to $5.5 \text{ Kg m}^{-2} \text{ h}^{-1}$ ($5.5 \text{ Kg m}^{-2} \text{ h}^{-1}$ for sea water). At the same time, a 12 hours' desalination test is conducted for the magnetic hierarchical assembly (Fig. R7a). Due to the tall 3D figure, the water supply is relatively restricted. Only the bottom part of the magnetic hierarchical assembly shows salt resistant feature, the taller parts are covered by precipitated salts after desalination (Fig. R7c). And the evaporation rate decreases from $5.4 \text{ Kg m}^{-2} \text{ h}^{-1}$ to $4.1 \text{ Kg m}^{-2} \text{ h}^{-1}$. However, only after water rinse or regeneration process, the magnetic hierarchical assembly could fully recover to its beginning figure (Fig. R7d) and performance (Fig. R7b).

7. In Figure 7, the magnetic hierarchical assembly leads more energy gaining from environment. The energy consumption in the system should be calculated. The explanations should be given to ensure the result. see the literatures *Adv. Funct. Mater.* **2019**, *29*, 1903255; *Joule*, **2018**, *2*, 1331-1338.

Reply:

According to the reported literature (*Adv. Funct. Mater.* **2019**, *29*, 1903255; *Joule*. **2018**, *2*, 1331-1338; *Joule*. **2019**, *3*, 1798-1803), the energy consumption in the solar steam generation systems can be divided into four parts, including water evaporation, the conduction to the bulk water, the conduction to the ambient air and infrared (IR) radiation. However, the last three parts can change to energy inflow in 3D evaporation

systems (*Joule*. **2018**, 2, 1171-1186; *Joule*. **2020**, 4, 928-937; *Adv. Sci.* **2021**, 8, 2002501), because the temperature of the evaporator could be lower than the environment. Note that the separate contributes to the environmental energy are hard to directly calculate because in our system the temperature distribution is nonuniform. Given the facts above, the energy consumption in the magnetic hierarchical assembly is only water evaporation. Hence, the energy input can be calculated from the evaporation rate, because the solar-vapor efficiency can be approximately regarded as 100% (*Adv. Sci.* **2018**, 5, 1800222).

Rotation rate (rpm)		0	25	50	75	100	125	150
Energy input (W m ⁻²)	Environmental energy	1572	1782	2014	2088	2122	2173	2190
	Solar-thermal power	1000	1000	1000	1000	1000	1000	1000
Energy consumption(W m ⁻²)	Water evaporation	2572	2782	3014	3088	3122	3173	3190

Table R1. The energy analysis in the magnetic hierarchical assembly during dynamic evaporation.

As shown in Table R1, under the illumination of 1 sun, the solar-thermal power is constant for different rotation rates, and can be regarded as being entirely consumed to water evaporation. Besides, the energy gain from the environment (conduction and IR radiation) keeps increasing as the rotation rate goes up, which is in accordance with the decreasing surface temperature of the evaporator (according to Fourier's Law and Stefan-Boltzmann equation, the heat flow positively correlates with the temperature difference). Besides, the in dark static evaporation rate of the magnetic hierarchical assembly is 2.78 Kg m⁻² h⁻¹ (1575 W m⁻²), this value approximately equals the environmental energy during static evaporation under 1 sun, which further proves the correctness of the energy analysis. These results have been added in the revised

supplementary information.

8. The evaporation enthalpies of water in Fe_3O_4 and $\text{Fe}_3\text{O}_4@\text{GO}$ are lower than that of bulk water, but that of water in $\text{Fe}_3\text{O}_4@\text{G}$ is higher, in Supplementary Figure 19, the author should give some explanations about the change in the evaporation enthalpies.

Reply:

We carefully reconsidered the measurement. Due to the limited accuracy of the Differential Scanning Calorimetry (DSC), the systematic errors must be taken into consideration. At the same time, during DSC test, the constant pressure heat capacity keeps changing as water keeps escaping from the system, which indeed deviates the basic line. In this regard, we recalculated the DSC results and conducted parallel experiments.

Figure R8. (a) DSC spectrum, (b) the calculated evaporation enthalpies of bulk water and contained water in Fe_3O_4 , $\text{Fe}_3\text{O}_4@\text{GO}$, and $\text{Fe}_3\text{O}_4@\text{G}$ nanoparticles.

As shown in Fig. R8b, the evaporation enthalpies of the contained water in Fe_3O_4 , $\text{Fe}_3\text{O}_4@\text{GO}$, and $\text{Fe}_3\text{O}_4@\text{G}$ nanoparticles are approximately the same, which are

slightly smaller than that of the bulk water. Hence, it can be considered that there is no significant difference between bulk water and the contained water. There is another possibility, the reduction of evaporation enthalpies of the contained water can be attributed to the intermediate water (*Sci. Adv.* **2019**, *5*, eaaw5484.) or water clusters during evaporation (*Nat. Nanotechnol.* **2018**, *13*, 489-495.). However, regarding the enthalpy decreases here are neglectable, we prefer to ascribe them to the systematic error of DSC test. Note that the evaporation enthalpy of water in Fe₃O₄@G nanoparticles affects the theoretical limit of evaporation. All the related data have been updated in the revised manuscript.

9. Line 207-211, “Under illumination of 0 – 1000 kW m⁻², the evaporation rate of CA assembly is higher than the theoretical limit, which is calculated from the evaporation enthalpy^{12,13,41} (Supplementary Fig. 19). However, it becomes lower than the theoretical limit when illumination larger than 1000 kW m⁻².” The author should check the expression of “1000 kW m⁻²”, the highest intensity in the text is only 2 sun illumination.

Reply:

It is a clerical error. The actual expression should be “0 – 1000 W m⁻²”, which has already been corrected in the revised manuscript.

Reviewer #3 (Remarks to the Author):

This work is devoted to an important area related to the development of new efficient devices for obtaining fresh and clean water. The method proposed by the authors refers to solar desalinators. Clean water can be obtained by evaporation of seawater or wastewater with its subsequent condensation. The use of solar energy is a budget-friendly and environmentally friendly approach, but, as a rule, inefficient. For this reason, the authors are investigating the possibility of increasing the evaporation rate by using a ferromagnetic liquid. A colloidal solution consisting of iron oxide nanoparticles and water is applied to the surface of the device. Graphene has been added to ensure the stability of the system, which prevents nanoparticles from agglomeration. Due to the influence of the magnetic field, conical structures and other forms of free surface are created. The black color of the solution contributes to a good absorption of solar radiation. In addition, a non-flat surface increases the evaporation area. An additional increase in the evaporation rate is given by an alternating magnetic field, which causes the dynamics of cone assemblies and other structures. As the water evaporates from the free surface of the structure, liquid flows from the container along a special path. The authors investigate the evaporation rate in a static and dynamic structure, how the rate of change of the magnetic field affects this process, the influence of the shape of the structure on the evaporation rate and in which areas evaporation is most intense. The authors also note that in this approach there is practically no crystallization of salt on the free surface, which is an additional advantage. A series of full-scale experiments and numerical calculations have been carried out in this work, which make it possible to deduce the main effects and mechanisms. The text of the

manuscript is written in a clear language. Of course, this work is important for the further development of this area. The lack of clean water is one of the world's biggest problems, especially for the countries of the African continent. But, nevertheless, before publishing this manuscript, it is necessary to deal with a number of problematic places, which I will mention below.

Thank you for your comprehensive and positive comments. Your constructive suggestions really help a lot to improve the manuscript. In the revised manuscript and supplementary information, numerical solution of the theoretical mechanism model is established. The graphical images and specific simulated results are clearer to elucidate the evaporation process. Questions about Monte Carlo are also answered point-to-point. Below are the details of the response to the comments.

Lines 58-61. Here you need to further explain. Due to what is the transportation of water? Filtration of liquid due to the capillary effect in a porous medium formed by voids between nanoparticles in a cluster?

Reply:

For the static evaporation, water is absorbed and transported by the capillary force, which is enabled by the abundant and cross-scale pores contained in the $\text{Fe}_3\text{O}_4@\text{G}$ CA assembly (Fig. R9a and b).

For the dynamic evaporation, the water transportation is believed to be contributed by two separate parts, which are capillary force caused by pore structure and the inner circulation caused by the rearrangement of the nanoparticles. As proved by the

luminescence tracing (Fig. R9c and d), the internal movement of the nanoparticles could carry with substances and render an inner circulation, which also accounts for the enhanced water supply.

Figure R9. Illustration for the origin of water transportation. **a, b** SEM images of the $\text{Fe}_3\text{O}_4@\text{G}$ assembly. **c, d** The luminescence tracing of fluorescein ions in the dynamic evaporation process.

Lines 232-234. In my opinion, the tip of the conical structure is relatively colder, because evaporation is more intense there, since the gradient of vapor concentration near the surface is higher in that area. After all, evaporation also occurs near the cone from the horizontal surface. To describe this mathematically, it is necessary to solve numerically not only the heat equation for the liquid phase, but also the equation of

vapor diffusion in the air region. In addition, the relatively rapid accumulation and crystallization of salt in the area of the tip of the conical structure additionally indicates that evaporation occurs more intensively in that area, respectively, the capillary flow transfers more dissolved matter to this area. What do you think about this?

Reply:

We share the same idea with you. However, our model and related discussions seem to be over simplified that fails to clarify the mechanism. Thus, complete equations of this system as well as its numerical solution is given in the revised manuscript, which is described as follows.

At the bevel of the cone, the thermal conduction differential equation can be expressed as

$$-\kappa \cdot \nabla^2 T = Q_{in-sun} + Q_{evap}, \quad (1)$$

Among them, κ is the heat conductivity coefficient for the assembly and the damp air, corresponding to $0.59 \text{ W} \cdot \text{m}^{-1} \cdot \text{K}^{-1}$ and $0.026 \text{ W} \cdot \text{m}^{-1} \cdot \text{K}^{-1}$ respectively. Q_{in-sun} is the solar-thermal heat source. Because the solar-thermal power is $1000 \text{ W} \cdot \text{m}^{-2}$ for the projected area, thus Q_{in-sun} for the bevel of the cone can be given by $Q_{in-sun} = 1000 \times \frac{2}{\sqrt{2^2+5^2}} \text{ W} \cdot \text{m}^{-2} = 371.39 \text{ W} \cdot \text{m}^{-2}$. And Q_{evap} is the heat consumption by water evaporation, and its value can be given by $Q_{evap} = -\Delta H_{evap} \cdot g_{evap}$, ΔH_{evap} is the enthalpy of evaporation which equals $40.8 \text{ kJ} \cdot \text{mol}^{-1}$, g_{evap} is the evaporation rate at the surface.

According to our proposed mechanism, the collective evaporation process can be divided into two consecutive processes: interfacial evaporation and water vapor

diffusion. The interfacial evaporation rate could be described by Dalton's evaporation equation, given by

$$g_{evap} = K \cdot (p^* - p), \quad (2)$$

where p^* is the saturated vapor pressure, and p is the vapor pressure near over the surface, K is the evaporation coefficient, equals $2.88 \times 10^{-7} s \cdot m^{-1}$ here, which is obtained from the experiment. And water vapor diffusion rate could be described by Fick's first law, which is

$$\mathbf{g}_w = -D \cdot \nabla P, \quad (3)$$

where \mathbf{g}_w refers to the diffusion flux of water vapor, D is the diffusion coefficient which is $2.6 \times 10^{-5} m^2 \cdot s^{-1}$ for 28 °C water vapor, ∇P is the water vapor pressure gradient near the surface. For evaporation process in a continuous and steady state, we could make steady-state approximation, i.e., p remains constant and g_{evap} equals the normal component of \mathbf{g}_w . Here we get

$$g_{evap} = \mathbf{n} \cdot \mathbf{g}_w, \quad (4)$$

where \mathbf{n} is the normal unit vector. Therefore, the governing equations in this system could be given by

$$-\kappa \cdot \nabla^2 T = 371.39 W \cdot m^{-2} - \Delta H_{evap} \cdot g_{evap}, \quad (5)$$

$$g_{evap} = K \cdot (p^* - p) = -D \cdot \mathbf{n} \cdot \nabla P, \quad (6)$$

and can be numerically solved by finite element method. In detail, the model is set to a cone with bottom radius = 2 mm, height = 5 mm, which is surrounded by cubic shaped damp air with side length = 30 mm. The relative humidity and temperature of the air boundary is set to 50% and 28°C respectively.

The corresponding numerical results are listed here. It is indeed that the evaporation at the top is much more intense, which is caused by the larger difference value between p^* and p (Δp). And the origin for this is the lower vapor pressure at the top. The relatively wide space at the top helps the vapor pressure escape (Fig. R10). In consequence, the higher evaporation rate at the top leads to more energy consumption, lowering the surface temperature, and in turn gaining more energy from the environment.

Figure R10. Plots of simulated water vapor pressure from evaporation surface of the top and the bottom to the surroundings, inset is plot of the simulated evaporation rate with respect to the height under 1 sun illumination.

The simulated tendency of the surface temperature distribution is also consistent with that of the experiment, which validates the correctness of our model (Fig. R11).

Figure R11. Numerical calculation results of the temperature distribution of the CA assembly under 1 and 2 sun illumination. (a) The experimental results and (b) the simulated results. (c, d) The simulated temperature distribution diagrams of the cone under 1 sun (c) and 2 sun (d) illumination.

Besides, the energy gained from the environment can also be given. As shown in Fig 12, the energy only flows in the evaporator when the surface temperature is lower than the environment. And the evaporation under 2 sun illumination only loses energy to the environment, which is in line with the fact that its evaporation rate is lower than the theoretical limit.

In short, the numerically solved model can reflect the basic physics of the evaporation process, i.e., the evaporation is mainly dominated by the vapor diffusion process, thus

the top evaporates faster than the bottom, leading to lower surface temperature, which is the critical point for the direction of the thermal flow. To gain more energy from the environment, it is necessary to promote the vapor diffusion process.

All of these discussions are updated in the revised manuscript and the revised supplementary information.

Figure R12. Plots of simulated energy gaining from the environment with respect to height of the CA assembly under 1 and 2 sun illumination.

Lines 248-249. Based on equation (1), it is difficult for me to understand why the cold tip evaporates faster. I have already expressed my opinion on this matter above. How does the theta angle affect the evaporation rate? In the Supplementary Information, eq. (4) is referred to as Fick's first law. In Fick's first law, there must be a derivative in space that is not in (4). How does the theta angle appear in (4)?

Reply:

1) To understand why the cold tip evaporates faster, we need to clarify the causality.

Here evaporation is the cause while the temperature variation is its consequence. At the tip, it has a relative larger space leading to the faster diffusion of the steam towards the environment associated with faster evaporation. As a result, more heat energy is taken away at the tip and thus a lower temperature could be expected.

2) Note that the complete form of the equations and the numerical solution have been updated in the revised manuscript and supplementary information.

Actually, as you mentioned, Fick's first law should involve the gradient part, which is substituted in our simplified model. To avoid misunderstandings, the original simplified model is abandoned, and the gradient part appears in Equation (R6). Secondly, in the original simplified model, the theta angle is introduced to describe the fact that vapor diffusion near the top is better than that of the bottom, which results from the conic shape of the evaporator. Below are the details about this point.

Considering the initial stage where the temperature is uniform everywhere. Given that the vapor source uniformly distributes on the surface of the cone, the equilibrium vapor concentration distribution is described by the corresponding Green function integration on the surface of the cone,

$$C(\mathbf{r}') = D \iint_{S_c} G(\mathbf{r}, \mathbf{r}') dS, \quad (7)$$

where $G(\mathbf{r}, \mathbf{r}') = \frac{1}{|\mathbf{r} - \mathbf{r}'|}$ (here Green's functions of 3D Laplace operator in the infinite space are applied), \mathbf{r} is the coordinate, D is a coefficient, and S_c represents the surface of the cone. Besides, H represents the total height of the cone, θ_0 is the angle between the cone axis and the normal of the cone surface, and R is the radius of the cone's bottom, which are marked in the schematic diagram of the cone geometry shown

in Fig. R13a.

To study the influence of the angle θ_0 to the vapor concentration, H is gradually changed with R being fixed ($R = 2.5$). As can be seen in Fig. R13b, smaller θ_0 leads to a more uniform vapor concentration distribution on the cone surface. Note that in Fig. R13b both the vapor concentration and height are normalized by their maximum values, the corresponding path is marked with red in Fig. R13a. In Fig. R13c and d, the vapor concentration distribution in $H = 1$ and $H = 5$ cases are given, which clearly shows smaller θ_0 (or lower H) leads to a more uniform vapor concentration distribution on the surface of the cone.

Since the vapor concentration distribution difference only originates from the geometry difference, our theoretical analysis confirms the critical point that more space at the tip leads to a much faster vapor diffusion and thus a lower vapor concentration. In this way, the theta angle appears in the simplified evaporation equation. However, this point can be also described more accurately by ∇P in Equation (R6), which is clearly reflected by the numerical solution (Fig R10). Note that new equations are adopted in the revised manuscript, in which there is no theta angle.

Figure R13. The influence of theta angle on the vapor diffusion in space. **a** The schematic diagram. **b** Plots of the vapor concentration with respect to the normalized height. From top down, the Height changed from 5 to 1, corresponding to θ_0 varying from 63.4° to 21.8° . **c, d** vapor concentration distribution in the space of the cone when $H = 5$ (**c**), and $H = 1$ (**d**).

Lines 260-262. “ii) disturbing the near atmosphere by dynamic evaporation, which is much more effective as it directly eliminates the water vapor gradient near the surface.”

Eliminate? Maybe, on the contrary, you need to increase the vapor gradient at the surface? After all, the higher the gradient, the more intense the evaporation occurs. I

think you should to remove the word "gradient".

Does this mean adding an air flow to the system in order to remove vapor even faster from the surface of the evaporator, thereby speeding up the process of liquid-vapor transition?

Reply:

1) Here, “eliminates the water vapor gradient near the surface” is a confusing expression, as you mentioned, it is the water vapor gradient ($p^* - p$) that dominates the evaporation rate (According to Dalton’s evaporation law, *Mem. Lit. Philos. Soc. Manchester*. **1798**, 5, 535-602; *Sci. Rep.* **2016**, 6, 34115). Thus, this sentence has been changed to “lowers the water vapor near the surface” in the revised manuscript.

2) The evaporation rate is increased by lowering p near the surface, which is the consequence of the dynamic evaporation process. However, this may result from two reasons. Partially, dynamic motion of the CA assembly causes air flow in the system to carry water vapor away from the surface. On the other hand, the dynamic motion could also expose CA assembly to the air with lower vapor pressure.

Fig. 4i. I believe that in this schematic Fig. 4i at the bottom there are not enough arrows indicating the evaporation flux density directed from the horizontal free surface on both sides of the cone. This is the key point in explaining that the vapor concentration gradient will be higher near the tip of the cone (see my comment for Lines 232-234).

Reply:

This multiple conic structure is a typical pattern in magnetic fluid called Rosensweig instability, where periodic cones emerge and pack closely. Plotting the horizontal free

surface could be misleading for readers to consider there is a large distance between these cones. On the other hand, the added model mentioned above is used to understand the observation that higher vapor concentration appears near the bottom of the cone.

Lines 339-340. "mean square radius of motion". Does this mean "mean squared displacement" (MSD)? It's a common term.

Reply:

The expression has been corrected in the revised manuscript.

Line 511. "2000 individual nanoparticles". With such a number of particles, the question immediately arises. Does the size of the cone structure of the model correspond to the size of the same structure from the experiment? What is the approximate size of nanoparticles? If the dimensions of the structures differ many times, then this point in the model needs to be further justified. It should be explained why, with a significant difference in scale, the model will describe the experiment.

Reply:

- 1) The size of the cone structure in the simulation is smaller than that of the experiment.
- 2) The size of the nanoparticles in the simulation is in accordance with that of the experiment, which is 100 nm (diameter).
- 3) According to the size of the cone, the particle number is about 6.4×10^{13} , which is impossible to simulate directly. Therefore, we choose to establish a miniature model to qualitatively (no quantitative comparison is involved) study the particles arrangement

during the rotation process. Note that in principle the size difference may only lead to a quantitative difference, but it cannot change the qualitative behavior (particles arrangement), since they share the same microscopic interaction as well as mechanism. By comparing slow and fast dynamics cases (see Supplementary Note 3), it is found that only when particles' rearrangement is sufficient, the cone structure could remain, otherwise the cone distorts gradually in the rotation process. Based on these simulation results, we infer the particles' rearrangement should be rapid in experiments as the cone structure is well kept during the rotation, which is also confirmed by element tracing experiments (see Fig. 6g in the revised manuscript).

Besides, this miniature model can be also considered as a small part of the cone located at the top tip, and the simulation results thus could reflect the particles' arrangement at this local place, which is beneficial for qualitatively understanding the inner circulation in the dynamic evaporation process.

Lines 512-513. Why does the model not take into account the presence of liquid in a porous medium and capillary flow? A mixture of particles and liquid is a solution. Doesn't the viscosity of the solution have an effect on this process? In the absence of a liquid, won't particles go away from this structure when exposed to a magnetic field?

Reply:

1) Monte Carlo simulation focuses on investigating the rearrangement process of the nanoparticles inside the cone. Since the size of water molecules are several orders smaller than the nano particle, they are not included directly in our simulation, only its

influence on particles is considered as two parts, i.e., the thermal disturbance and some modification to the inter-particle interaction. This approximation is widely used in modelling. Our simulation follows the standard way to colloidal systems (*An introduction to dynamics of colloids*, Elsevier, New York ; Amsterdam, **1996**, p250-300. ; *Colloidal Dispersions*, Cambridge **1992**, p260.) and complex fluids (*Phys. Rep.* **2004**, 390, 453-551; *The theory of polymer dynamics*. Oxford University Press, Oxford, **1986**, p281; *Dynamics of polymeric liquids*, 2nd ed., Wiley, New York, **1987**, p260.), the interacting many-particle system is treated on a mesoscopic level, which means the microscopic details of the particles and solvent is neglected (*Colloidal Magnetic Fluids : Basics, Development and Application of Ferrofluids*, Springer, **2009**).

2) The viscosity influences the diffusion of nanoparticles. From the perspective of energy, higher viscosity is corresponding to higher energy barrier when particles moving, which can be reflected by the current interactions in the simulation.

3) As the potentials used in our simulations correctly describe the microscopic interactions, the cone structure and its configuration evolution under varying magnetic field is well reproduced. For this reason, the cone structure is relatively stable and doesn't collapse during the simulation process, since the assembly state is more energy favorable. Both the inter particles potential and magnetic dipole interaction can contribute attractions between two nanoparticles and help particles to form an assembly. Though occasionally several particles can run away from the assembly, it doesn't affect studying the particles' rearrangements inside the cone structure.

Line 536. A table should be added indicating all the parameters of the problem, their values and units of measurement.

Reply:

A table containing all necessary parameters used in the Monte Carlo simulation is depicted (Table R2), and has been added in the revised supplementary information.

Here, the scaled value is in accordance with the experimental results but all normalized with the same proportion in the simulation. And the corresponding typical energy is in consistence with the experiments.

Potential	Typical energy	Physical quantity	Scaled value in simulation
Lennard Jones potential	3.37×10^{-19} J	ε	0.003
		σ	1
Magnetic dipole interaction	3.29×10^{-18} J	$\mu_0/4\pi r_{ij}^3$	0.03
Magnetic force	7.26×10^{-17} J	B_z	2
		$B_x^2 + B_y^2$	1
Gravitational potential	2.54×10^{-16} J	mg	0.15
Thermal effect	3.89×10^{-18} J	$K_B T$	0.0345

Table R2. The table of parameters in Monte Carlo simulation.

Lines 537-539. There is no description of the calculation algorithm here. It is necessary to provide at least a brief description or in the form of a pseudocode. Only formulas of particle interaction energies are indicated. How was the particle dynamics simulated and the particle collision processed? What is the essence of the Monte Carlo method used here? For what quantities was a random number generator used? Which random number generator was involved? Has an existing package been used or has a computer program been developed?

Reply:

1) We have added a logic diagram in the revised supplementary information. As shown in Table R3, our method follows the standard Monte Carlo simulation steps (*A guide to Monte Carlo simulations in statistical physics*, Fifth edition. ed., Cambridge University Press, Cambridge, **2021**).

Firstly, the initial configuration (including coordinates and the dipole moments of particles) is set to a pre-relaxed tilted cone consisting of 2000 particles (the stable configuration in the initial external magnetic field). In each step, every particle in the system undergoes a possible configuration change process. Specifically, for each particle, the displacement is randomly generated with its amplitude no more than a given value. By comparing the energy of the new and current configuration, a transition state condition is judged to determine whether this particle movement could happen or not. After every particle accomplishing this process, the program changes the orientation of the external magnetic field and runs into the next Monte Carlo step until the end (step = MC_step).

Table R3. The logic diagram of the Monte Carlo simulation.

2) Monte Carlo mainly focuses on the evolution of particles' position rather than particles' dynamics, while their velocity effect is taken into account by thermal energy. All interactions are described by potentials, which defines the energy for a given configuration of particles. In Monte Carlo method, the collision between two particles leads to energy arising. As a result, these two particles tend to separate.

3) Monte Carlo method is a computational algorithm that relies on repeated random sampling to obtain numerical results. The underlying concept is to use randomness to solve problems that are deterministic in principle. The necessity of using Monte Carlo method here comes from its merit of solving problems with many coupled degrees of freedom. Hence, by employing Monte Carlo method, the simulation system can be simplified and the computation amount can be reduced. Meanwhile, based on important sampling analysis using statistical thermodynamics, the investigation is not restricted

by the time-scale.

4) With setting a seed, pseudorandom numbers are generated uniformly from 0 to 1, whose statistical randomness could be guaranteed. This random number generation method is widely used in random sampling, Monte Carlo methods, board games, or gambling (<https://en.wikipedia.org/wiki/Pseudorandomness>).

5) Although there is some package existing, we choose to develop a computer program, which is exclusively designed and optimized for our Monte Carlo simulation.

REVIEWERS' COMMENTS

Reviewer #2 (Remarks to the Author):

All comments from reviewers have been addressed carefully. I recommend it for publication without further review.

Reviewer #3 (Remarks to the Author):

The article has been significantly revised. The authors have answered all the questions. The manuscript is now ready for publication in NC.